



# Modelling extensive green roof CO2 exchanges in the TEB urban canopy model

Aurélien Mirebeau[1], Cécile de Munck[1], Bertrand Bonan[1], Christine Delire[1], Aude Lemonsu[1], Valéry Masson[1], and Stephan Weber[2]

[1]Centre National de Recherches Météorologiques (CNRM), Université de Toulouse, Météo-France, CNRS, Toulouse, France
[2]Technische Universität Braunschweig (TUBS), Institute of Geoecology, Braunschweig, Germany

**Correspondence:** Aurélien Mirebeau (aurelien.mirebeau@meteo.fr)

**Abstract.** Green roofs are promoted to provide ecosystem services and to mitigate climate change in urban areas. This is largely due to their supposed benefits for biodiversity, rainwater management, evaporative cooling, and carbon sequestration. One scientific challenge is quantifying the various contributions of green roofs using reliable methods. In this context, the green roof module already running in the Town Energy Balance urban canopy model for water and energy exchanges was improved
by implementing the CO2 fluxes and the carbon sequestration potential. This parametrisation combines the ISBA (Interaction Between Soil Biosphere and Atmosphere) photosynthesis, biomass and soil respiration module with the green roof module in order to quantify the net CO2 amount emitted or fixed by the green roof over a time period. The parametrisation was fully achieved by using data of an extensive Sedum non irrigated green roof located at the Berlin BER airport in Germany from 2016 to 2020. The five years of measurements were used to do a sensitivity analysis of the photosynthesis module parameters
in order to classify the parameters according to their influence, followed by a calibration over the most important parameters and evaluation. Results show a good agreement of the simulated leaf area index and CO2 fluxes with *in situ* observations, with good diurnal, seasonal and inter-annual variability, even if the model tends to be overly responsive on the day to day variability. The model reproduces well the Net Ecosystem Exchange which provides a reliable estimation of the annual carbon sequestration. Those results are encouraging in quantifying the potential of carbon sequestration of green roofs and open up
the possibility of applying the new parametrisation on a city-wide scale to evaluate green roof scenarios.

## 1 Introduction

Green roofs refer to roofs with a vegetated surface on top of a growing layer. They are mainly divided into two categories: extensive green roofs, which are made with shallow substrate, low-profile plant species without necessarily irrigation, and the intensive green roofs which can support shrubs and trees and require irrigation and deeper substrate. Recent studies have
investigated the various advantages of both green roofs types. This covers issues including their impact on building energy savings at building scale (Virk et al., 2015) and at city scale (Wang et al., 2024) under different climates (Ascione et al., 2013), on urban water management, especially the effects on runoff quantity (Zheng et al., 2021) and runoff quality (Li and Babcock, 2014), but also their interaction with plants, animals, and the abiotic environment (Cook-Patton and Bauerle, 2012)





and their benefit on air quality (Currie and Bass, 2008). A further advantage is the reduction of carbon emissions, in two ways

as pointed out by Tan et al. (2023). Firstly, an indirect reduction is due to less carbon being emitted as a result of building energy savings. In addition, a direct reduction comes from the carbon sequestration by the soil and vegetation on the green roof. In this perceptive, recent studies, like the work of Seyedabadi et al. (2021), estimated the direct carbon sequestration in containers with dry weight measurement for *Sedum acre*, *Frankenia thymifolia*, and *Vinca* major species at 38, 565, and 166 g C m$^{-2}$ yr$^{-1}$, respectively. In addition, the indirect carbon sequestration was quantified by modelling at 7680, 7222, and 6393

g C m$^{-2}$ yr$^{-1}$, respectively for the same species. Kuronuma et al. (2018) estimated, also with dry weight measurements, the direct carbon sequestration by an extensive green roof for irrigated *Cynodon dactylon*, *Festuca arundinacea*, *Zoysia matrella* irrigated *Sedum aizoon* and non irrigated *Sedum aizoon* at 690, 751, 671, 459, and 336 g C m$^{-2}$ yr$^{-1}$, respectively. They also evaluated the payback of time for non irrigated extensive *Sedum aizoon* green roof ranging between 8.5 and 14.0 years, hinting that CO2 sequestration could be a real positive effect of green roof. However, as highlighted by Shafique et al. (2020), most of

the studies quantifying the potential carbon sequestration rely on short term observations.

In addition, in order to really determine the feasibility and interest of such installations, impact studies must be conducted considering all interactions on a city-wide scale and over long enough time periods to cover a variety of climatic conditions. This illustrates the relevance of representing green roofs fluxes, including CO2 exchanges, in appropriate urban land surface model, such as the Town Energy Balance (TEB, Masson, 2000) model, that can be run in various configurations: with or

without a complex atmospheric coupling, from the street level to the neighborhood, the city or regional-scale, and for specific meteorological events or seasonal even multi-annual time periods.

The TEB model already includes an extensive green roof module for addressing heat, energy and water exchanges (de Munck et al., 2013). The model has already been applied in the Paris region to quantify the benefits of green roofs in terms of urban cooling, improved thermal comfort, and energy savings (de Munck et al., 2018). We propose here to improve the existing model

by adding the calculation of CO2 fluxes for green roofs. The aim of this article is to provide a full description and evaluation of the added CO2 fluxes in the TEB's extensive green roof module in order to have a model able to quantify the CO2 sequestration of green roofs with its annual and inter annual variations. This is done by reusing the model called Interaction Soil-Biosphere-Atmosphere (ISBA, Noilhan and Planton, 1989; Noilhan and Mahfouf, 1996) which represents CO2 fluxes of the soil and vegetation. But in order to fit with the specificity of green roofs (shallow soil and Sedum vegetation), a new parametrisation of

ISBA is achieved with calibration and evaluation on experimental data collected for several years over an extensive green roof with Sedum species on top of an airport car park in Berlin.

The experimental data are presented in Sect.2. Subsequently, the TEB model, the TEB-GREENROOF module, and the implementation of CO2 fluxes are developed in Sect.3, followed by the description of the model configuration and numerical setup for the study case in Sec. 4. Then, the sensitivity analysis, calibration and evaluation of the new TEB-GREENROOF

version including CO2 fluxes with the observational data are presented in Sect. 5. Finally, Sect. 6 is a discussion about the behavior of the Sedum simulated with the new parametrisation, the diurnal cycle of CO2 fluxes, and the quantification of the amount of carbon fixed by the green roof.





## 2 Instrumented green roof for model development and evaluation

The modelling is informed and evaluated by comparison with continuous observations collected on an experimental green roof

site for several years by the Technische Universität Braunschweig (Heusinger and Weber, 2017a, b; Konopka et al., 2021). The site is a non-irrigated extensive green roof of 8600 m² (see picture Fig. 1) constructed in May 2012. It is located on the flat roof of a 18 m high car park at Berlin Brandenburg airport in Germany (referred to as BER, 52.37°N, 13.51°E, altitude of 61 m above sea level). The green roof is composed of four layers: (1) a vegetation layer made up mainly of sedum (*Sedum floriferum 'Weihenstephaner Gold, Sedum album'*) with herbaceous plants (*Allium schoenoprasum, Trifolium sp*), (2) a 0.09 m

deep substrate layer composed of a mix of expanded shale, pumice and compost, (3) a 0.003 m thick protection mat, and (4) a 0.05 m thick insulation layer. It is supported on a 1.60 cm layer of ferroconcrete. Gardeners provide basic maintenance of the roof vegetation, approximately once a year.

The site is equipped with a 3D ultrasonic anemometer and an open-path infrared gas analyser at 1.15 m above roof level to determine the net CO2 fluxes, and the turbulent latent (LE) and sensible (H) heat fluxes by the eddy-covariance technique.

The site is also equipped with radiometers to measure both downwelling and upwelling components of long-wave (LW$^{\downarrow}$, LW$^{\uparrow}$) and short-wave (SW$^{\downarrow}$, SW$^{\uparrow}$) radiation. Air temperature and humidity are measured at 2 m above the green roof surface by an HMP155 probe and precipitation data are gathered from the nearby German Weather Service station at Berlin Schönefeld airport (BSCH; ID: 00427). Probes are also placed in the substrate layer to measure the soil temperature and water content at different depths (0.025, 0.05, and 0.075 m). The fraction of vegetation cover ($F_{veg}$) and the leaf area index (LAI) are estimated

occasionally ($\simeq$10 times a year) through photograph analysis on 10 different locations randomly selected on the roof.

The site has been in operation since mid-2014. This study focuses on the period 2016-2020, for which a very comprehensive dataset is available. According to the Köppen climate classification (Köppen, 1900), Berlin is located in a region of temperate oceanic climate (Cfb). Winters are cold and summers are warm and humid. The rainfall pattern indicates moderate rainfall throughout the year (annual average of 591 mm according to Lorenz et al., 2019), with climatological maxima in June and

July. Snowfall is typical from December to March. Figure 2 presents the monthly values for the daily minimum and maximum temperatures (TN and TX, respectively), and precipitation for the period of interest. The five years selected show contrasted meteorological conditions with a noticeable inter-annual variability in temperature and precipitation. 2018 was a particularly dry year, with total rainfall well below normal (only 66 mm recorded in summer, compared to 157 mm on summer average for all the period), and also slightly warmer (average summer TX of 26.0°C, compared to 25.0°C on average for all the period).

2019 is also slightly drier and, above all, warmer than average (120 mm of precipitation and average TX of 26.4°C in summer). In contrast, 2017 is wetter and colder in summer (268 mm of precipitation and an average TX of 23.5°C).



**Figure 1.** Photograph of the green roof experimental plot located on top of the car park of the Berlin Brandenburg airprot (Germany).

## 3 Model description and implementation

### 3.1 Overview of the TEB urban canopy model

The Town Energy Balance (TEB) model is integrated in the open-access SURFEX land surface modelling plateform (Masson
et al., 2013) together with other dedicated surface models (for vegetation and natural soils, lakes, oceans etc.).

The TEB model is a surface scheme developed by Masson (2000) to represent heat, water, and momentum exchanges between urban surfaces and the atmosphere, and to compute street-level microclimatic conditions. TEB can be run at street or city scale, with meteorological forcings or coupled to an atmospheric model due to its simplified geometry.





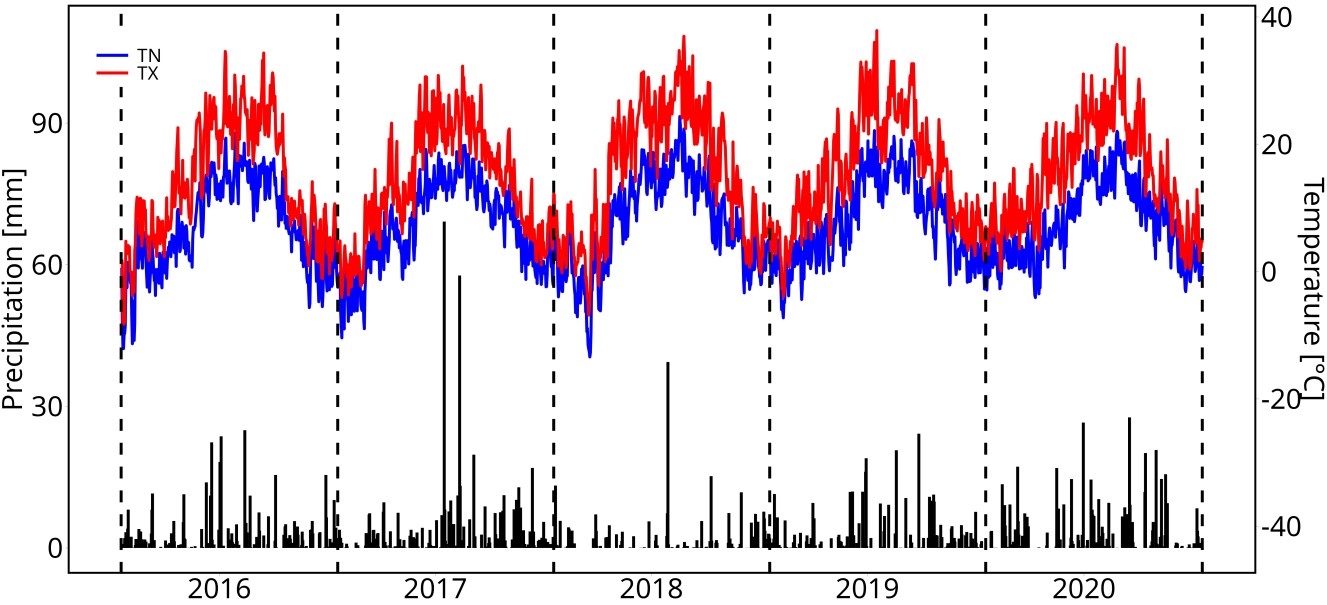

**Figure 2.** Evolution of meteorological conditions over the period 2016-2020: daily maximal temperature (TX, in °C), daily minimal temperature (TN, in °C), and daily cumulative precipitation (black bars, in mm)

In TEB, the urban geometry is represented by the concept of street canyon (Oke, 1987). This hypothesis considers that
urban areas can be roughly represented as a single road between two facing buildings of same height and infinite length, and
with flat roofs. With this geometry, the model computes the radiation, energy and water balances on each surface of the street
canyon (wall, road, and roof) and aggregates the fluxes to simulate the exchanges between the overall urban canopy layer and
the atmosphere above. To better describe the heterogeneity of the urban environment in TEB, the interactions between urban
vegetation (i.e. ground vegetation and street trees) and built up surfaces are now represented within the street canyon (Lemonsu
et al., 2012; Redon et al., 2017, 2020). To model the functioning of urban vegetation, TEB relies on the soil and vegetation
surface scheme Interaction Soil-Biosphere-Atmosphere (ISBA, Noilhan and Planton, 1989; Noilhan and Mahfouf, 1996) that
is also integrated in the SURFEX platform.

### 3.2    Initial version of the TEB-GREENROOF module

In addition, the TEB-GREENROOF module (de Munck et al., 2013) was developed in TEB to allow for the simulation of exten-
sive green roofs. It is also based on the physics of ISBA in order to describe the soil and vegetation layers of the green roof and
model its hydrological and thermal performances in interaction with the building on which it is installed, as well as exchanges
with the atmosphere above. Figure 3 shows the different fluxes simulated by the original version of TEB-GREENROOF for
the hydrological and thermal processes (panels a and b). At surface level, the thermal balance is estimated according to the





equation:

$$c_s \frac{\partial T_S}{\partial t} = Q* + H - LE - G_0 \tag{1}$$

Where $Q_*$ is the net radiation (W m$^{-2}$), $H$ is the sensible heat flux (W m$^{-2}$), $LE$ is the latent heat flux (W m$^{-2}$), $G_0$ is the surface ground heat flux (W m$^{-2}$), $c_s$ the surface soil heat capacity (J K$^{-1}$ m$^{-3}$) and $T_s$ is the surface temperature (K).

The thermal balance connects to the water balance through the latent heat flux ($LE$). At the surface, the latent heat flux is the sum of the plant transpiration ($LE_{TR}$), the soil evaporation ($LE_G$), and the evaporation of the water intercepted by the leaves ($LE_V$). This last term is estimated using a water interception reservoir for leaves that is supplied with a set fraction of the precipitation. The transpiration ($LE_{TR}$) is calculated with the following ISBA parametrisation:

$$LE_{TR} = F_{veg} \cdot \rho_a C_H V_a H_V VPD \tag{2}$$

where $F_{veg}$ is the fraction of vegetation, $\rho_a$ the air density (kg m$^{-3}$), $C_H$ the turbulent exchange coefficient (W m$^{-2}$ K$^{-1}$), $V_a$ the wind speed (m s$^{-1}$), $VPD$ the vapour pressure deficit in the air (kPa), and $H_V$ the Halstead coefficient (dimensionless). The Halstead coefficient depends in particular on the LAI which will be modelled later with the implementation of CO2 fluxes.

The soil of the green roof is described using the multi-layer diffusion version ISBA-DF (Boone et al., 2000; Decharme et al., 2011), which allows for the discretisation of the soil into different vertical layers. In each soil layer, the evolution equation of soil temperature follows the expression:

$$c_g \frac{\partial T}{\partial t} = \frac{\partial G}{\partial z} + \phi \tag{3}$$

where $T$ is the soil temperature ($K$), $G$ is the vertical ground heat flux (J m$^{-2}$ s$^{-1}$), $c_g$ is the soil heat capacity (J K$^{-1}$ m$^{-3}$), $\phi$ is a latent heat source/sink resulting from phase transformation of soil water (J m$^{-3}$ s$^{-1}$), and $z$ the soil depth (m).

For the evolution of soil liquid water and soil ice volumetric content (m$^3$ m$^{-3}$), the equations are:

$$\frac{\partial w_l}{\partial t} = -\frac{\partial F}{\partial z} - \frac{\phi}{L_f \rho_w} - \frac{S_l}{\rho_w} \quad (w_{min} \leq w_l \leq w_{sat}) \tag{4}$$

$$\frac{\partial w_i}{\partial t} = \frac{\phi}{L_f \rho_w} \quad (0 \leq w_i \leq w_{sat} - w_{min}) \tag{5}$$

where $F$ is the vertical water flux (m s$^{-1}$), $L_f$ is the latent heat of water fusion (J kg$^{-1}$), $\rho_w$ is the liquid water density (kg m$^{-3}$), and $S_l$ represents external sources/sinks for liquid water (kg m$^{-3}$ s$^{-1}$).

In the soil, the hydrological and thermal balances are coupled through the effective thermal properties of each soil layer $j$, which change over time as a function of the soil water content. This is the case for both the layer-averaged soil heat capacity ($c_{gj}$, in J K$^{-1}$ m$^{-3}$) and thermal conductivity ($\lambda_j$, in kg m s$^{-3}$ K$^{-1}$) for layer $j$, according to:

$$c_{gj} = (1 - w_{satj})c_{dryj} + w_{lj}c_l + w_{ij}c_i \tag{6}$$





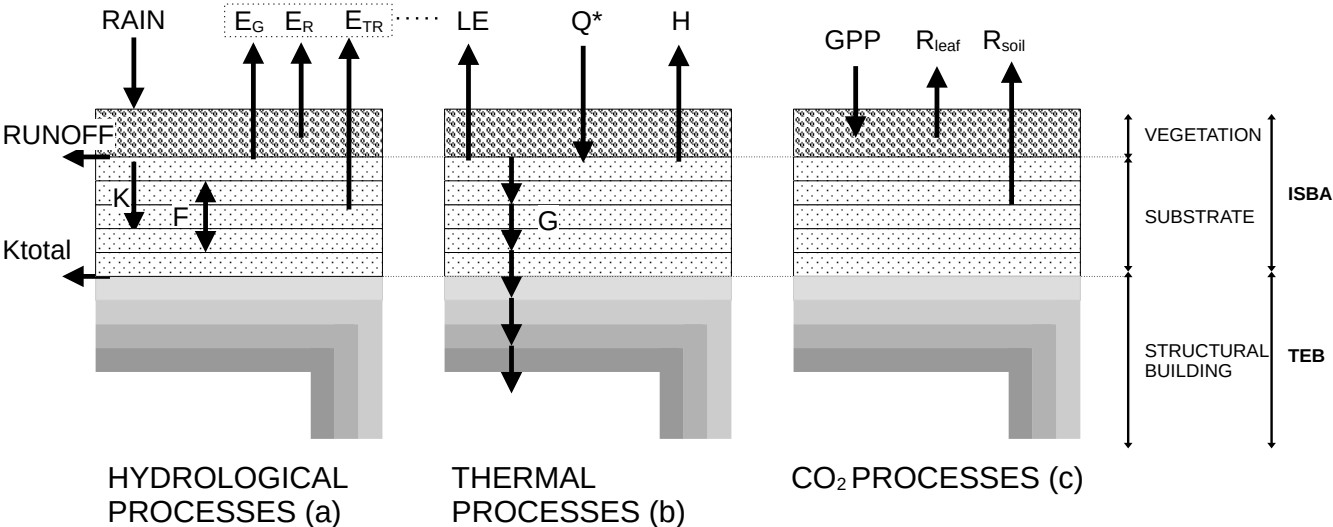

**Figure 3.** (a) TEB-GREENROOF hydrological processes: water gain from precipitation (RAIN), surface runoff(RUNOFF), drainage (K), total drainage (Ktotal), vertical water fluxes (F), ground evaporation $E_G$, evaporation of water intercepted by vegetation $E_R$, and vegetation transpiration $E_{TR}$, all in $\mathrm{kg\,m^{-2}\,s^{-1}}$ ; (b) TEB-GREENROOF thermal processes : net radiation (Q*), latent heat flux (LE), sensible heat flux (H) and ground heat flux (G), all in $\mathrm{W\,m^{-2}}$.(c) TEB-GREENROOF CO2 processes : growth primary production by photosynthesis (GPP), leaf respiration ($R_{leaf}$) and soil respiration ($R_{soil}$)

$$\lambda_j = K_e \lambda_{sat\,j} + (1 - K_e)\lambda_{dry\,j} \tag{7}$$

where $w_{lj}$ and $w_{ij}$ are the soil liquid water content and soil ice content in layer $j$ (m³ m⁻³). $c_{dry\,j}$ is the heat capacity of
140  dry soil matrix for layer $j$, and $c_l$ and $c_i$ are the heat capacities of liquid water and ice, respectively. $K_e$ is the dimensionless Kersten number, and $\lambda_{sat\,j}$ and $\lambda_{dry\,j}$ are the saturated and dry soil thermal conductivities for layer $j$, respectively.

### 3.3 Implementation of CO2 fluxes in TEB-GREENROOF

As with the modelling of radiative, thermal and hydrological exchanges in TEB-GREENROOF, the CO2 fluxes are modelled in TEB-GREENROOF by activating, with some adaptations, an existing module of ISBA designed to represent C fluxes in
145  vegetation and soils.





The main fluxes are plant photosynthesis (or gross primary production referred to as $GPP$, in g C m$^{-2}$ yr$^{-1}$) and ecosystem respiration ($R_{ECO}$, in g C m$^{-2}$ yr$^{-1}$) which combines leaf respiration ($R_{leaf}$, in g C m$^{-2}$ yr$^{-1}$) and soil heterotrophic respiration ($R_{soil}$, in g C m$^{-2}$ yr$^{-1}$) (see Fig. 3c)

The difference between these two large fluxes is defined as the Net Ecosystem Exchange ($R_{eco} = R_{leaf} + R_{soil}$) (Bonan, 2016).

Photosynthesis and leaf respiration are modelled with the A-gs parametrisation (Calvet et al., 1998, 2004) that is based on the assimilation scheme proposed by Jacobs (Jacobs, 1994; Goudriaan et al., 1985) (see Appendix A). This semi-empirical model has the advantage of being simple and suited for both C3 and C4 plants. The approach chosen considers that light and CO2 atmospheric concentration are the two limiting factors impacting the net photosynthetic rate.

The goal of this study is to adapt some aspects of the present parametrisation in order to model Berlin's airport extensive green roof by representing the behaviour of the dominant Sedum species (which is widely used for extensive green roofs). Sedum are facultative C3-CAM (Winter, 2019), meaning that they behave like a C3 plant when they are well watered and switch to a CAM photosynthesis pathway when they lack water. With a CAM behaviour, the stomata of the plant open only at night to fix CO2. The photosynthesis takes place during the day, requiring light energy, but the leaf stomata remain closed to prevent water loss through evapotranspiration. ISBA, like most large-scale vegetation models, does not represent the photosynthesis mechanisms specific to this particular species and a few adaptations are needed to take their behaviour into account. On the Berlin's airport extensive green roof, the observations did not show clear evidence of CAM behaviour with CO2 absorption during night and not during day, so it was decided not to investigate further into the modelling of this aspect. However, it still requires adaptation of the model with a specific parametrisation to match the behaviour of Sedum on a shallow substrate.

First, an analysis of the observed CO2 fluxes measured by eddy covariance did not show any high and low temperature inhibition of photosynthesis in the range of temperatures observed. Because we could not find data on high and low temperature inhibition for these species, we decided to put the inhibition functions to 1 (see Appendix eq. A2 and A3) for the maximum net CO2 assimilation ($A_{m,max}$, in g m$^{-2}$ s$^{-1}$) and the mesophyll conductance ($g^*_{mes}$, in m s$^{-1}$):

$$A_{m,max}(T_s) = A_{m,max}(25) \cdot Q_{10}^{\frac{T_s - 25}{10}} \tag{8}$$

$$g^*_{mes}(T_s) = g^*_{mes}(25) \cdot Q_{10}^{\frac{T_s - 25}{10}} \tag{9}$$

where $A_{m,max}(25)$ is the maximum net CO2 assimilation at 25 °C in mg m$^{-2}$ s$^{-1}$, $T_s$ is the leaf skin temperature (K), here considered as the first layer of soil temperature, and $g^*_{mes}(25)$ is the mesophyll conductance at 25 °C.

When soil moisture content drops, plants tend to close their stomata to limit water loss. This is described empirically by a soil water stress function ($F2$) that simply model the stomata closing and opening when the plant lacks water according to a regulation of the mesophyll conductance:

$$g_{mes} = g^*_{mes}(T_s) \cdot F2 \tag{10}$$





with the normalized soil water stress factor estimated as:

$$F2 = \sum_{j=1}^{N} [\Upsilon_j \cdot (\frac{w_j - w_{wilt\,j}}{w_{fc\,j} - w_{wilt\,j}})] \tag{11}$$

where $N$ is the number of soil layers, and $\Upsilon_k$ the root fraction in layer $j$. $w_{wilt\,j}$ is the wilting point, $w_{fc\,j}$ the field capacity, and $w_j$ the soil water content in layer $j$. As the substrate of extensive green roof is shallow, thresholds $F2_{min}$ and $F2_{max}$ are prescribed for $F2$ to prevent photosynthesis cut or too excessive variations. This empirical formulation is simpler than the one proposed by Calvet (2000) for herbaceous plants.

The respiration of the soil (in kg m$^{-2}$ s$^{-1}$) is estimated with the simple Norman et al. (1992) respiration scheme:

$$R_{soil} = 4.4 \cdot 10^{-8} \cdot (13.5 + 5.4 \cdot LAI).w_{10} \cdot e^{0.069 \cdot (T_{s10} - 25)} \tag{12}$$

where LAI is the leaf area index (m$^2$ m$^{-2}$), $w_{10}$ is the weighted soil volumetric water content between 0-10 cm depth (%) and $T_{s10}$ is weighted soil temperature between 0-10 cm depth (°C). Since the soil on green roof is very shallow, water content thresholds $w_{10_{min}}$ and $w_{10_{max}}$ are set to prevent extreme values.

Furthermore, the A-gs CO2 assimilation scheme can be either forced by a prescribed LAI or coupled to a biomass scheme
making the LAI evolve over time. We use the ISBA NIT (Calvet and Soussana, 2001) biomass scheme that only represents above-ground biomass, separated into three different reservoirs: the leaf biomass reservoir, the above ground stem biomass reservoir, and a residual reservoir.

Each biomass reservoir $B_i$ (expressed in kg of dry matter (DM) m$^{-2}$) follows the same time evolution equation:

$$\frac{\Delta B_i}{\Delta t} = A_i - D_i - R_i \tag{13}$$

with $A_i$ the biomass input, $D_i$ and $R_i$ respectively the biomass loss due to mortality, allocation to other reservoirs and respiration. The input flux for leaf biomass is the carbon assimilation due to photosynthesis (expressed in kg C m$^{-2}$ d$^{-1}$):

$$A_{leaf} = A_{n,day} \cdot \Delta t \tag{14}$$

After a daily biomass iteration, the LAI is calculated from the leaf biomass and the specific leaf area (SLA, in m$^2$ kg DM$^{-1}$) as follows:

$$LAI = SLAI \cdot B_{leaf} \tag{15}$$

Except for crops, ISBA assumes a constant vegetation cover with time. But extensive green roof vegetation can be patchy and the vegetation cover may vary in time, which is the case on the experimental site studied here. To take this heterogeneity and variability into account, the fraction of vegetation cover is estimated by fitting the equation on estimation of LAI and vegetation cover:

$$F_{veg} = 1 - e^{a \cdot LAI} \tag{16}$$





where $a$ is a coefficient set on the basis of observations.

All the vegetation parameters required for the CO2 flux modelling are listed in Table 2. Standard values are available in ISBA for each of these parameters, for both C3 and C4 plants (Gibelin et al., 2006). For application to green roofs, the main challenge is to find values specific to Sedum that are currently not described in ISBA as mentioned earlier. Since we could not
find appropriate ecophysiological data to derive these parameter values, we chose to calibrate some of them (see Sect. 5).

## 4 Configuration of TEB simulation

### 4.1 Atmospheric conditions

In this study, TEB is run in an offline simulation configuration over five years, from 2016 to 2021. It is applied to a single grid point, i.e. for an average urban canyon whose characteristics are representative of the study site, in particular the properties
of the green roof. In offline mode, the time evolution of atmospheric conditions over the urban canyon must be provided to the TEB model at both given altitude and time step. The data required are the above urban canopy air temperature, humidity, pressure and CO2 concentration, the wind speed and direction, precipitation and the short-wave (direct and scattered) and long-wave incoming radiations.

Here, the meteorological forcing is built with local green roof observations described in Sect. 2. and provided to the model
with a time resolution of 30 minutes. The specific humidity is calculated with the relative humidity, air pressure and the air temperature measurements. The partitioning of the incoming global short wave radiation into scattered and direct components is made based on the method of Erbs et al. (1982). Solid and liquid precipitation rates are determined by disaggregating the total precipitation, first according to the rain/snow precipitation classification directly provided from the nearby German Weather Service station at Berlin Schönefeld airport (BSCH; ID: 00427). When classification data are missing or indicate rain and snow
during the same day, disaggregation is done using the air temperature recorded on the roof by applying a threshold of 273.15 K.

### 4.2 TEB model configuration

The TEB model is run on a 1D grid, with the urban canyon dimensions and properties defined as the average parameters of the measuring site. The height of building is set to 18 m consistently with the airport parking lot characteristics. The forcing
height is determined according to the sensors location on the green roof, i.e. 19.15 m above ground level (agl) for wind speed and direction and 20 m agl for air temperature and humidity. TEB is running with a simplified surface-boundary-layer scheme (Hamdi and Masson, 2008; Masson and Seity, 2009) allowing for the vertical discretization of the atmosphere in the canyon (from the ground to the forcing height) into 6 layers of micro-climate variables. The fraction of building is set to 0.6 and fully covered by green roofs using the TEB-GREENROOF module, the fraction of road is 0.2, and the fraction of low vegetation
is 0.2 with GARDEN module (Lemonsu et al., 2012). The road is discretized into 2 layers corresponding to a surface layer of artificial coating and a basement layer corresponding to natural soil. The parking lot is a concrete building with large openings





on the facades. The walls are defined in the model with 2 layers of concrete, and the roof consists in a surface layer of concrete and a second layer of insulation. The TEB building energy module (Bueno et al., 2012; Pigeon et al., 2014) is also activated with adaptation specifically for the case study in order to represent the inside of the car park and the potential interactions
between the indoor and outdoor environment . All input parameters including the geometric, radiative and thermal properties of the TEB model are listed in Table B1 of Appendix B.

Besides the photosynthesis parameters (described in Tab. 2), the input parameters of TEB-GREENROOF module, including the description of the green roof and the thermal and hydrological properties, are described in Table 1. The green roof soil is discretized into 6 layers with same composition for a total depth of 0.09 m. With regard to hydrological properties, the potential
of the soil matrix at saturation is defined based on *ex situ* analysis. The empirical coefficient for water retention curve ($B_{coef}$) is set based on the results of a previous case study referenced in de Munck et al. (2013). The hydraulic conductivity at saturation is based on the soil manufacturer's documentation. The soil porosity profile, field capacity, and wilting point are set directly based on measurements of soil water content on the green roof. The definition of the thermal properties is based on the initial calibration proposed by de Munck et al. (2013) for an other instrumented extensive green roof: the same value of dry soil heat
capacity is applied but the dry soil thermal conductivity is slightly increased to better modelled the heat conduction in the soil according to measurements (not presented here). The vegetation albedo and emissivity are taken from the previous study of de Munck et al. (2013).

## 5   Definition of green roof CO2 fluxes parameters

Without information on the specific characteristics of green roofs, the input parameters for the photosynthesis model must be
defined. The initialisation of these parameters is done following three successive steps:

(1) Sensitivity analysis on the main parameters of the A-gs photosynthesis model. This step is based on a stand-alone and very low computing time version of A-gs (detailed in Appendix B). Forced by predefined microclimate conditions, a very large number of simulations are carried out to test a wide range of parameter values and clarify which parameters are the most significant and have the greatest effect on the calculation of CO2 fluxes.

(2) Pre-calibration that consists in running additional simulations, again using the stand-alone version of A-gs, but with a focus on the parameters selected in the previous step in order to narrow the range of plausible values (Appendix D).

(3) Calibration of the selected parameters and according to the values identified in the previous step. This time, several combinations of plausible parameter values are tested by running the full TEB configuration considering dynamic LAI. Based on these simulations, the best configuration can be identified.





**Table 1.** TEB-GREENROOF input parameters for green roof substrate and vegetation

| Type | Parameter | Unit | Values |
|---|---|---|---|
| **Geometry** | Numbers of numerical soil layer | (-) | 6 |
| | Layer thickness (from top to bottom) | cm | (0.3, 1.9, 1.9, 1.9, 1.5, 1.5) |
| **Surface** | Albedo of bare soil | (-) | 0.154 |
| | Emissivity of bare soil | (-) | 0.83 |
| | Albedo of vegetation | (-) | 0.2 |
| | Emissivity of vegetation | (-) | 0.83 |
| | Roughness length for momentum | m | 0.03 |
| **Thermal properties** | Dry soil thermal conductivity | $W\ m^{-1}\ K^{-1}$ | 0.15 |
| | Dry soil heat capacity | $J\ m^{-3}\ K^{-1}$ | 2000000 |
| **Hydrological properties** | Wilting point | $m^3\ m^{-3}$ | 0.001 |
| | Field capacity | $m^3\ m^{-3}$ | 0.20 |
| | Porosity profile | $m^3\ m^{-3}$ | 0.57 |
| | Matrix potential at saturation | m | -0.1 |
| | $B_{coef}$ coefficient for water retention curve | (-) | 4.0 |
| | Hydraulic conductivity at saturation | $m\ s^{-1}$ | $2.183\cdot10^{-3}$ |

## 5.1 Sensitivity analysis of ISBA-A-gs parameters

### 5.1.1 Sobol index method

In order to assess which ISBA A-gs parameters are the most predominant in modelling the NEE, a global sensitivity analysis is performed on the parameters listed in Table 2. The approach used is the Sobol method (Sobol, 1993, 2001) that can be applied for either linear or non-linear models. Two indices are considered here. The first one $S_i$ is the first-order global sensitivity of output $Y$ to a sole selected input parameter $X_i$, i.e. the variance of $Y$ when $X_i$ is the only parameter to vary. $S_i$ is normalized by the variance of $Y$ to obtain a score between 0 (no sensitivity) and 1 (full sensitivity).

The second index considered $S_{T_i}$ also known as the total-order index, calculates the variance of output $Y$ to $X_i$ when $X_i$ varies with the other parameters in every possible combination ($X_i$ varies solely, then $X_i$ varies with each parameter, then three parameters including $X_i$ vary together, ...). As $S_i$, $S_{T_i}$ is normalized by the variance of $Y$ with the score between 0 (no sensitivity) and 1.

To compute those two indices for each A-gs parameter, we use a Monte Carlo approach developed by Saltelli et al. (2010). This involves working with samplings that span adequately the space of possible values for each parameter. In this study, the sampling is performed with a latin hypercube sampling (Stein, 1987) that is a stratified sampling method aiming to spread





the sample points evenly across all possible values. This is done using the R package 'lhs' (Carnell, 2022) with the method
geneticLHS (Stocki, 2005) for a sampling with a genetic algorithm to maximize the mean distance from each point to all the
other points.

### 5.1.2  Application to the photosynthesis A-gs model

To implement this sensitivity study on the input parameters of the A-gs photosynthesis model, a simplified modelling configu-
ration is developed. The sensitivity analysis is carried out with the stand-alone A-gs model (Appendix B) on scores of simulated
CO2 fluxes. Mean absolute error ( MAE) and root-mean-square error (RMSE) are computed over the 5 years of observation
selected at a temporal scale of 30 minutes. Among all input parameters of the A-gs model, $d$=8 parameters are not set on
observation and are selected for the sensitivity analysis. The range of values for part of the parameters is based on a previous
sensitivity analysis from (Aouade et al., 2020) with reference values from the literature: $g_{mes}^{*}(25)$ from Calvet (2000), $g_c$ from
Gibelin et al. (2006), $D_{max}$ from Calvet (2000). For the other parameters, the ranges are set according to the default values
defined for types C3 and C4 in ISBA A-gs, with a $\pm$ 10 % margin: $\epsilon_0$, $\Gamma(25)$, $A_{m,max}(25)$ from Gibelin et al. (2006) and
$f_0^{*}$ from Jacobs (1994). The parameters and associated value ranges are listed in the Table 2. The distribution within the fixed
range of each parameter is defined according to a uniform distribution.

The size of each sampling matrix is set to $N = 5000$ which implies a total of $N \cdot (2 + d) = 50000$ simulations. Here, the
output variables of interest for which the Sobol indices are calculated are the mean absolute error (MAE) and the root-mean-
square error (RMSE) of the net CO2 fluxes simulated by A-gs compared with the net CO2 flux measurements collected on the
instrumented green roof.

### 5.1.3  Sensitivity analysis results

The results of the sensitivity analysis on the A-gs input parameters for its implementation in the TEB-GREENROOF model
for CO2 calculation are presented on Fig. 4 for both MAE and RMSE. The values obtained for the Sobol index make possible
to first identify the parameters with no influence on the net carbon fluxes scores, i.e. $D_{max}$ and $g_c$. As the result, they are
set to default values as listed in Table 2. All the other parameters are retained for the calibration with the same range for the
pre-calibration. Among them, $g_{mes}^{*}(25)$ is the most sensitive one, followed by $\Gamma(25)$ and $\epsilon_0$, with similar results for MAE
and RMSE. As a result, these three input parameters have the greatest influence on the performance of the A-gs model for
calculating CO2 fluxes, and thus require more meticulous calibration (that is presented in the next section). The parameters
$A_{m,max}(25)$, $f_0^{*}$ and $F2_{max}$ have a smaller impact but not negligible so that they also retained for the pre-calibration.

### 5.2  Calibration of A-gs parameters

### 5.2.1  Method

The model is calibrated, based on the observation dataset of net CO2 fluxes divided into two distinct time periods: a four-year
calibration period (from 2016 to 2019) and a one-year evaluation period (2020). A pre-calibration (Appendix D) is made to





**Figure 4.** Comparison of the Sobol first order (red) and total order (blue) indexes calculated for the eight variables in the sensitivity analysis, for both RMSE (left bar) and MAE (right bar).





reduce the range of values for the six photosynthesis parameters to be calibrated ($g^*_{mes}(25)$, $\Gamma(25)$ $\epsilon_0$, $A_{m,max}(25)$, $f^*_0$ and $F2_{max}$).

An ensemble of TEB-GREENROOF simulations is then run, according to the full configuration presented in Sect. 4.2 (and considering a dynamic LAI). Each simulation represents a different combination of parameter values. For each of the six photosynthesis parameters, two values are tested, based on the pre-calibration results Tab. 2. Since TEB-GREENROOF simulations are performed with a dynamic LAI, additional parameters related to the biomass evolution need to be calibrated, i.e. the maximal lifespan of leaves ($\tau_M$) and the specific leaf area ($SLA$). For *Sedum Album*, the value of $SLA$ is set to 12.9 m$^2$ kg DM$^{-1}$ according to the TRY database (Kattge et al., 2020) that brings together the different plant trait databases worldwide (TRY last version contains more than million trait records for 6.24 million individual plants). The parameter $\tau_M$ is tested for the two values of 75 and 150 days. In total, 128 combinations of parameters are simulated with TEB-GREENROOF. To identify the best configuration, the statistical scores of the 128 experiments calculated over 2016-2019 are compared, by crossing the mean absolute error (MAE) of net CO2 fluxes and the root-mean square error (RMSE) of LAI.

### 5.2.2 Calibration results

Figures 5 illustrates the outcomes of all simulations in terms of MAE for the net CO2 fluxes and RMSE for LAI, with details of the values of each parameter in all simulations. In accordance with the findings of the sensitivity analysis, the mesophyll conductance at 25 ° C is identified as the most influential parameter. On Fig. 5 (c) nearly all simulations with $g^*_{mes}(25)$=0.001 m s$^{-1}$ demonstrate better performances than simulations with $g^*_{mes}(25)$=0.002 m s$^{-1}$ for both net CO2 fluxes and LAI. The maximal lifespan of leaves ($\tau_M$) is also found to have a significant impact. However, as evidenced by the two distinct curves (Fig. 5 (a) ), this parameter primarily affects the representation of LAI, with systematic better RMSE for $\tau_M$=75 days than 150 days . For $\Gamma(25)$ and $\epsilon_0$ (Fig. 5 (d,f)), the two values tested for each parameter give quite comparable performances. It is nonetheless possible to identify a single value, considered as the best, set to 55 ppmv for $\Gamma(25)$ and 1.4 kg CO2 PAR for $\epsilon_0$. For the other parameters, namely $A_{m,max}(25)$, $f^*_0$, and $F2_{max}$ (Fig. 5 (b,e,g)), it is more difficult to conclude on the values leading to the best configuration. Indeed, the differences in performances between the simulations performed with the two values of $A_{m,max}(25)$ are very low, which corresponds well with the sensitivity analysis results. For $f^*_0$ and $F2_{max}$, the simulations with the best RMSE for LAI do not correspond to the simulations with the best MAE for CO2 fluxes, So a choice needs to be made. The best configuration is highlighted on Fig. 5 (a), with the corresponding parameter values listed in Table 2. It is selected because it gives the best simulation for LAI and still close to the best simulation for $F_{CO_2}$ MAE.

### 5.3 Evaluation of the TEB-GREENROOF

### 5.3.1 Dynamic LAI modelling

The modelling of the evolving LAI using the biomass model is presented in Fig. 6 and compared with the LAI retrieved from observations. Since observation-based LAI is estimated from the analysis of discontinuous photographs, it is difficult to assess the accuracy of the model in detail. Nevertheless, some interesting results stand out from the comparison. The temporal





**Figure 5.** Comparison of the performances of the 128 simulations combining the RMSE of daily LAI (x-axis) and the MAE of 30-min average instantaneous net CO2 fluxes (y-axis). Panel a highlights the simulation selected as the best (black dot) compared to the others (grey dots). For each other panel corresponding to one of the variables to be calibrated (b-h), the red/blue colour code distinguishes the performance of the two values tested for this variable.



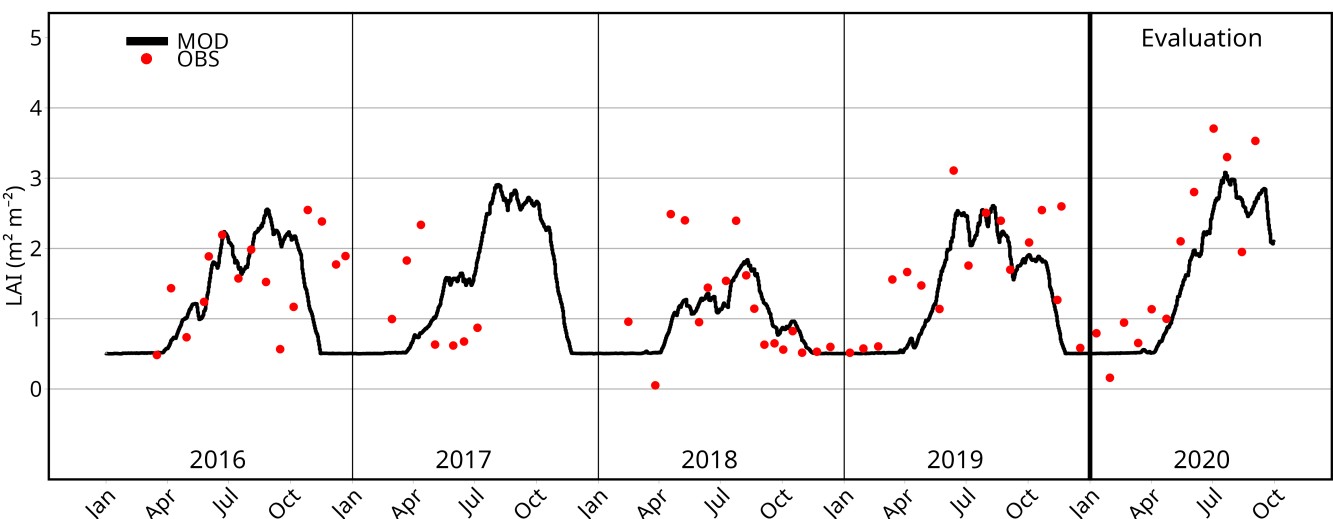

**Figure 6.** Daily evolution of the modelled LAI (black line) during the calibration (2016-2019) and evaluation (2020) periods, compared to the estimated on site LAI values (red dots), in $m^3$ $m^{-3}$.

evolution of the LAI reveals a good representation of the overall seasonal dynamics for the evaluation year 2020, with a clear identification of the growing period starting around April and ending in September. During this period, the annual maximum LAI is found in July, reaching 3.08 $m^2$ $m^{-2}$ for modelling and 3.71 $m^2$ $m^{-2}$ for estimation. For the calibration period, the

modelled annual maximum LAI reaches 2.56, 1.84, and 2.61 $m^2$ $m^{-2}$ for 2016, 2018, and 2019, respectively, which corresponds with the variation of annual maximum observed LAI at 2.55, 2.49 and, 3.11 $m^2$ $m^{-2}$ for the same years. Note that comparison cannot be done for year 2017 since there is no estimation of LAI after mid July. With regard to the senescence period, a minimum threshold of LAI is prescribed in the model at 0.5 $m^2$ $m^{-2}$ based on the estimated LAI between 2016-2020 (in particular, according to winter values in 2018-2019, see Fig. 6), which still matches reasonably well with what is estimated

in 2020. Finally, it is noteworthy that the model is able to simulate inter-annual variability in LAI. For the particularly dry year 2018, the average LAI simulated during the growing season (April-September) is much weaker (0.9 $m^2$ $m^{-2}$) than that of other years (1.6-1.7 $m^2$ $m^{-2}$). This value is in agreement with the LAI of 1.13 $m^2$ $m^{-2}$ estimated in 2018.

### 5.3.2 Net CO2 fluxes modelling

The monthly evolution of the net CO2 fluxes diurnal cycle simulated and observed over the year 2020 is represented on

Fig. 7. The model provides a good representation of the amplitude of the net CO2 fluxes for the evaluation period. Both the annual cycle and diurnal cycle are close to the observations. The model is able to capture the net CO2 flux seasonal variations, in response to variation of climatic forcing. In August and late April, both the model and the observations catch less

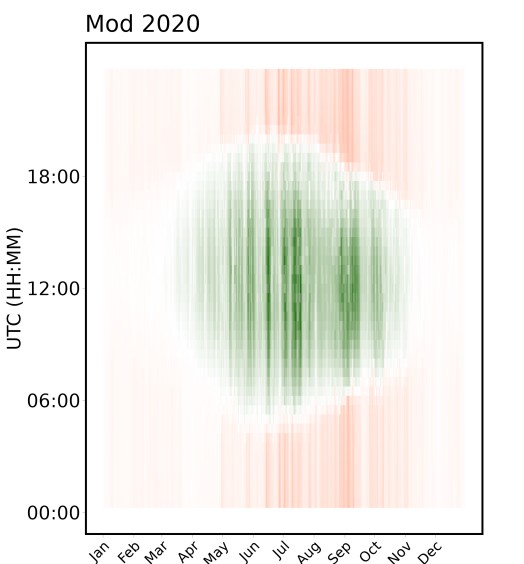
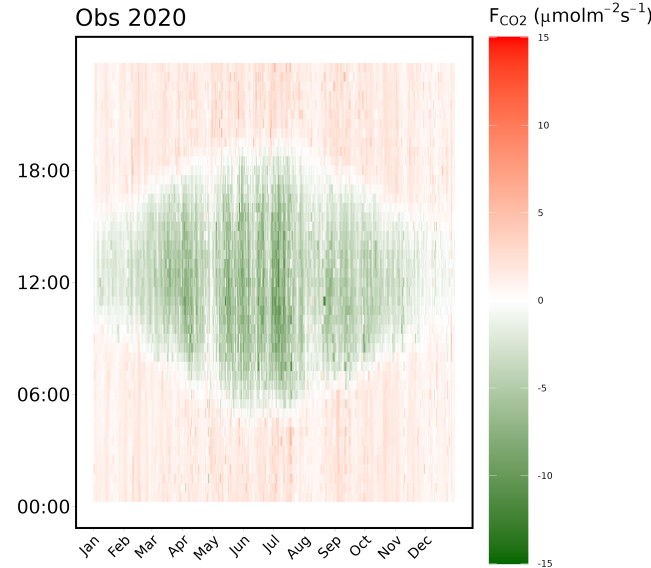

**Figure 7.** Monthly evolution of daily variation of modelled (left) and observed (right) CO2 fluxes ($F_{CO_2}$) for the evaluation period 2020. The colour range varies from green for negative CO2 fluxes (photosynthesis) to red for positive CO2 fluxes (respiration).

photosynthesis due to a dryer period of stress for the vegetation. Conversely, the model also reproduces periods in July when photosynthesis is enhanced. However, the model remains excessively responsive to meteorological fluctuations, particularly following precipitation events, when soil water content is high, leading to an overestimation of the photosynthesis. Furthermore, outside the growing season, the photosynthesis simulated by the model is close to zero, whereas observations indicate that photosynthesis continues even during winter.

## 6 Discussion

The work presented in the previous section has enabled us to characterize for the first time the parameters of the photosynthesis and vegetation growth model for Sedum in a Soil-Vegetation-Atmosphere-Transfer model. We now discuss in more detail the behaviour of the net CO2 fluxes from green roofs in the TEB model.

### 6.1 Sedum parameters

The calibration proposed in this study allows for the representation of Sedum behaviour when they do not use a CAM photosynthesis pathway. Here we investigate the response of the new Sedum parametrisation to the environmental variables driving photosynthesis (soil temperature, water content and radiation) compared to the standard C3 and C4 parametrisations. Figure 8



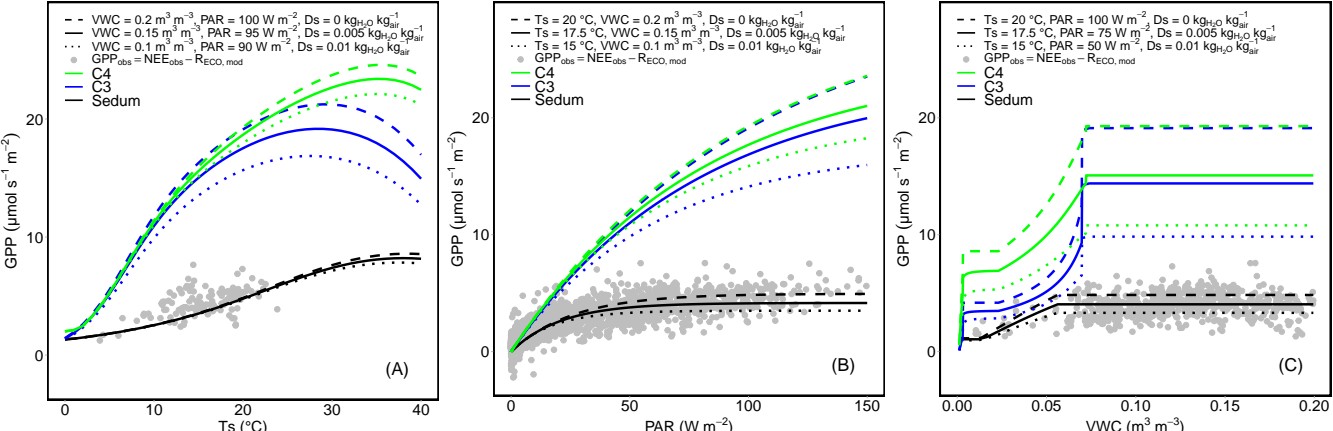

**Figure 8.** Evolution of the GPP with temperature (a) in °C, photosynthetically active radiation (b) in $W\ m^{-2}$ and volumetric water content (c) in m$^3$ m$^{-3}$, for the Sedum ISBA C3 and ISBA C4 parametrisation and for observation. For each parametrisation, three response curves are presented corresponding to the real environmental conditions observed on the site. In each panel, the three curves displayed for each parametrisation frame the real environmental conditions observed on the site. For comparison, the observations are selected within the range of the most extreme values of the three curves (the dashed and dotted lines). The observed GPP displayed on the three figures is estimated from the observation of net CO2 fluxes on the BER green roof site subtracted by the modelled $R_{ECO}$ of the best simulation. All observations were conducted within the specified range of each figure, with variables influencing photosynthesis set between the most and least favourable values for photosynthesis.

shows on each panel the response curves of GPP to the three environmental variables: $T_s$, $PAR$ and $VWC$. For each response curve to one variable, the other environmental variables are fixed. In each panel, the three curves displayed for each parametrisation frame the real environmental conditions observed on the site. For comparison, the observations are selected within the range of the most extreme values of the three curves (the dashed and dotted lines).

For the three curves, the Sedum parametrisation performs better than the ISBA C3 and ISBA C4 parametrisations with a significantly lower photosynthetic rate. The response to volumetric water content (Fig. 8c) demonstrates that the threshold values for the normalized water stress factor ($F2_{min}$ and $F2_{max}$) and the weighted soil volumetric water content between 0-10 cm depth for soil respiration ($w_{10_{min}}$ and $w_{10_{max}}$ ) were appropriately selected, this is visible on the plateau reached at a $VWC$ of 0.056 m$^3$ m$^{-3}$ on the average micro-meteorological conditions (straight line) which correspond well with the 380 observations. With regard to the photosynthetically active radiation (Fig. 8b), the Sedum parametrisation models correctly represent the increase of GPP but also the threshold above which the GPP no longer increase with photosynthetically active radiation. This response is not noted with the ISBA C3 and ISBA C4 parametrisations. For the response to temperature (Fig. 8a), the range of temperatures observed does not make possible the determination of the maximum GPP achievable at higher temperatures, and beyond which the GPP decreases with temperature.




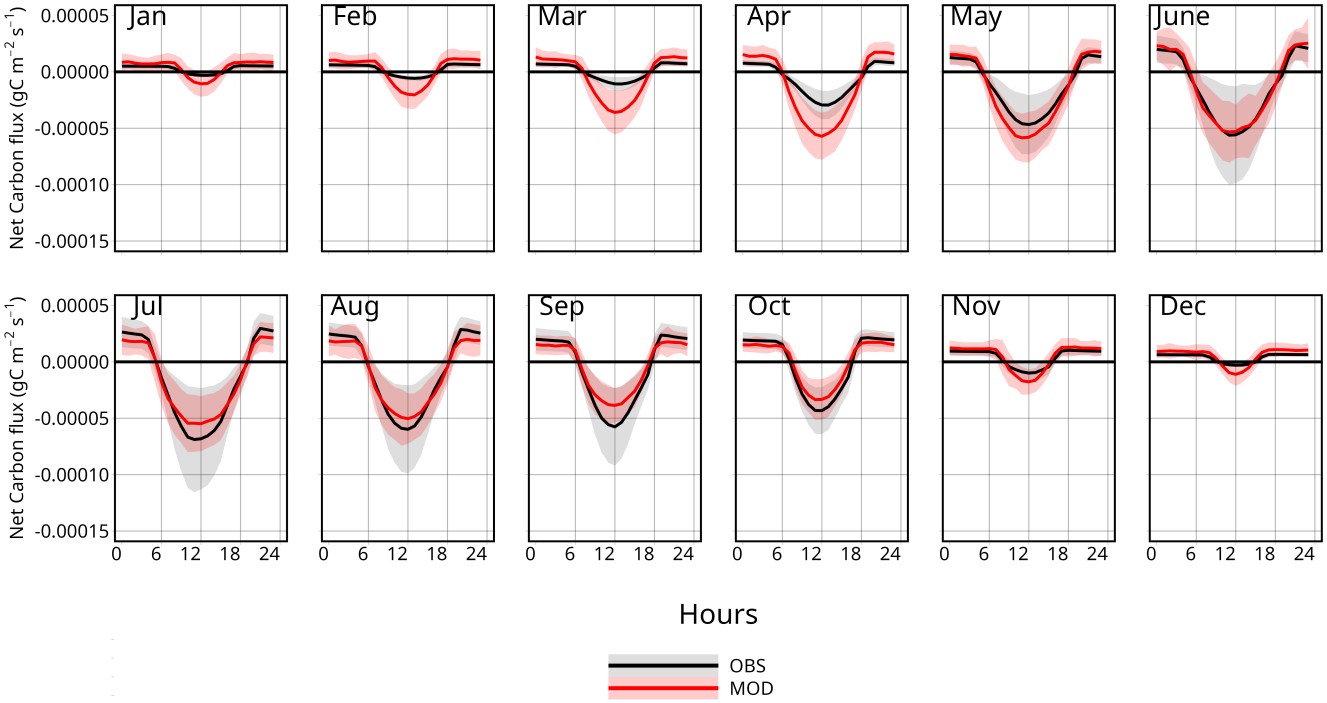

**Figure 9.** Comparison of the modelled (black line) and observed (red line) diurnal cycles of net CO2 fluxes (g s$^{-1}$ m$^{-2}$) monthly averaged over the five-year period. The transparent ranges indicate standard deviations.

## 6.2 Diurnal cycle

The diurnal cycle of the net CO2 fluxes monthly averaged over the five-year period is illustrated in Fig. 9. Similarly to what was shown in Fig. 7, outside the growth period (between November and March), the model does not simulate the diurnal CO2 cycle, which is still noticeable in the observations and reflects weak but still active photosynthesis. Howerver, the amplitude of the net CO2 fluxes is on average quite well represented during the growing period, especially between June and October. During the day, the CO2 fluxes are well represented, although the model tends to be overly responsive compared to the observations, resulting in a greater standard deviation for the modelling. The modelled respiration at night is in close agreement with the measurements as it can be seen between 8 pm and 6 am, and follows well the annual variation, being greater during the growing period and close to zero outside the growing period.

## 6.3 Net ecosystem exchanges

The net ecosystem exchange (NEE) is calculated over a given time period, over which it makes it possible to quantify the net CO2 sequestration if the NEE is negative, or the net CO2 emissions if the NEE is positive. Figure 10 represents the observed and modelled daily NEE monthly averaged for the five successive years from 2015 to 2020. The modelled NEE is in close



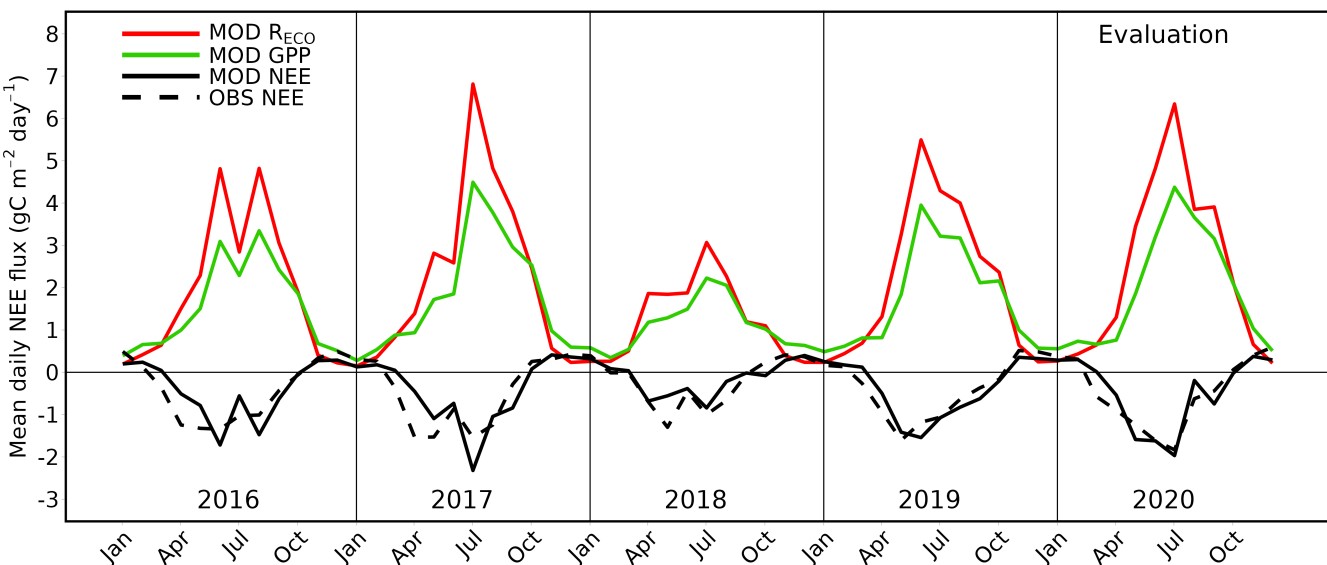

**Figure 10.** Comparison of the modelled (black line) and observed (black dashed line) monthly averaged daily NEE (in g C m$^{-2}$ day$^{-1}$) for the period 2016-2020. For simulation only, the monthly averaged of both daily ecosytem respiration (red line) and daily GPP (green) are also presented.

agreement with the measurements for all years. It follows the observed annual variation, with a slightly positive NEE from October to March, and negative NEE for the rest of the year. This means that the green roof acts as a net carbon emitter in

autumn and winter, but as a net carbon sink in spring and summer. The partitioning on the modelling between $GPP$ and $R_{ECO}$ on Fig 10 demonstrates that both $GPP$ and $R_{ECO}$ follow similar inter-annual variations, being greater in summer than in winter. However since no partitioning are available between $GPP$ and $R_{ECO}$ on observations during this study, it is not possible to confirm the accuracy of the model processes. At the annual scale (Table 3), the model shows that the green roof is a net carbon sink in accordance with measurements, and quantifies fairly well the amount of carbon fixed by the green roof

within the range of error of the measurement estimated at 16 g C m$^{-2}$ yr$^{-1}$. Indeed, the observed and modelled annual NEE for the evaluation year 2020 is -168 and -166 g C m$^{-2}$ yr$^{-1}$, respectively. The model is also able to capture the inter-annual variations in NEE, which are partly governed by changes in weather conditions: especially, the year 2018, which stands out as a dry year compared with normals, shows a lower annual NEE than the other years in both observations and simulation (-85 and -51 g C m$^{-2}$ yr$^{-1}$, respectively). Inversely, the wetter year 2017 presents a greater observed NEE of -178 g C m$^{-2}$ yr$^{-1}$)

than the other years, that is not directly reflected in the simulation even though the NEE is rather high (-163 g C m$^{-2}$ yr$^{-1}$).



# 7 Conclusions

A new parametrisation for the net CO2 fluxes calculation has been implemented in the model TEB-GREENROOF by using the ISBA-A-gs photosynthesis model with a biomass module and an empirical parametrisation of the ecosystem respiration. The modelling was informed by observations on an extensive non irrigated Sedum green roof located at the Berlin BER airport.

The sensitivity analysis results showed that the main parameters driving the CO2 fluxes on the green roofs are the mesophyll conductance at 25°C ($g^*_{mes}(25)$), the CO2 compensation concentration at 25°C ($\Gamma(25)$), and the maximum initial quantum use efficiency ($\epsilon_0$), which needed to be calibrated more carefully. The results after calibration showed that the model performed well in reproducing the diurnal cycle during the growing period and its dependence to the variations in meteorological conditions and soil water condition. Outside the growing period, the model struggles in simulating the weak photosynthesis process that

seems to persist based on observations. Nonetheless, as CO2 fluxes remain very low during this period, the impact on the overall quantification of carbon sequestration is limited, leading to a good estimation of the annual NEE and of its interannual variations. This work also ultimately allowed to characterise for the first time the photosynthesis and growth parameters appropriate for modelling Sedum with the ISBA SVAT model.

Future development will need to include comparison between the ecophysiological parameters calibrated here and on site

measurements of green roof plants photosynthesis parameters. To study the full carbon cycle of the green roof, the management and maintenance of green roofs need to be addressed, especially by considering the biomass that can be removed by gardeners. Similarly, the carbon sequestered in the soil needs to be quantified with on site sampling. On the modelling side, the growth module needs to be improved, and a soil organic carbon module needs to be added in order to be able to perform longer term studies.

Along side thermal and hydrological effect, the short term carbon sequestration can be now added into impact studies in order to have a full picture of the impact of green roof on the fluxes at city scale and under different climate events and conditions. But the green roof effect should not be only looked in term of fluxes, other effect that cannot be modelled in model like SURFEX such as effect on individuals and biodiversity should also be taken into account, thus require to cross-reference the results of different approaches.

*Code and data availability.* TEB is part of the software SURFEX from the CNRM open source website: https://opensource. umr-cnrm.fr under the CeCILL Free Software License Agreement v1.0 license. The version with net CO2 fluxes modelling for green roof is available on 10.5281/zenodo.14289462. The experimental data that are used for the calibration and evaluation were provided by Prof. Stephan Weber from the Technische Universität Braunschweig, it is necessary to contact Porf. Stephan Weber directly.





**Appendix A:**

**A-gs equation:** The following equations are use in the ISBA-A-gs in SURFEX V9. The model uses an empirical light response function of net assimilation ($A_n$) to combine the effects of light and CO2 as limiting factors. When light is not limiting, $A_m$ is limited by a maximum photosynthetic rate $A_{m,max}$:

$$A_m = A_{m,max}[1 - e^{-g^*_{mes}\frac{(C_i - \Gamma)}{A_{m,max}}}] \tag{A1}$$

where $C_i$ is the internal leaf CO2 concentration in kgCO2 kgair$^{-1}$. $A_{m,max}$ is the maximum net CO2 assimilation in mg m$^{-2}$

s$^{-1}$, $g^*_{mes}$ the mesophyll conductance in m s$^{-1}$, and $\Gamma$ the CO2 concentration compensation point in ppmv, defined according to the following equations:

$$A_{m,max}(T_s) = \frac{A_{m,max}(25) \cdot Q_{10}^{\frac{T_s - 25}{10}}}{(1 + e^{0.3(T_1 - T_s)}) \cdot (1 + e^{0.3(T_s - T_2)})} \tag{A2}$$

$$g^*_{mes}(T_s) = \frac{g*_{mes}(25) \cdot Q_{10}^{\frac{T_s - 25}{10}}}{(1 + e^{0.3(T_1 - T_s)}) \cdot (1 + e^{0.3(T_s - T_2)})} \tag{A3}$$


$$\Gamma(T_s) = \Gamma(25) \cdot Q_{10}^{\frac{T_s - 25}{10}} \tag{A4}$$

where $A_{m,max}(25)$ is the maximum net CO2 assimilation at 25 °C in mg m$^{-2}$ s$^{-1}$, $g^*_{mes}(25)$ is the mesophyll conductance at 25 degree in m s$^{-1}$, $T_s$ is the leaf skin temperature in °C, $T_1$ and $T_2$ are reference temperatures in °C.

The internal CO2 concentration depends directly of the atmospheric CO2 concentration (Eq. A5) and is controlled by the air

humidity (Eq. A6).

$$C_i = f \cdot C_s + (1 - f)\Gamma \tag{A5}$$

where $C_s$ is the atmospheric CO2 concentration in kgCO2 kgair$^{-1}$ and $f$ a coupling factor estimated via:

$$f = f_0^* \cdot (1 - \frac{D_s}{D_{max}^*}) + f_{min} \cdot (\frac{D_s}{D_{max}^*}) \tag{A6}$$

where $D_{max}^*$ is the maximum specific humidity deficit of the air tolerated by vegetation in kg$_{H_2O}$ kg$_{air}^{-1}$, $D_s$ is the leaf to air

saturation deficit in kg$_{H_2O}$ kg$_{air}^{-1}$ and $f_0^*$ is the value of the $f$ factor for $D_s$=0 g kg$^{-1}$ and $f_{min}$ given by:

$$f_{min} = \frac{g_c}{g_c + g^*_{mes}} \tag{A7}$$

where $g_c$ is the cutilcular conductance in m s$^{-1}$.





Then the CO2 assimilation is limited by the photosynthetically active radiation via:

$$A_n = (A_m + R_d) \cdot (1 - e^{-\frac{\epsilon * I_a}{A_m + R_d}}) - R_d \qquad (A8)$$

where $\epsilon$ is the initial quantum use efficiency in $kg_{CO2}$ $J^{-1}$ PAR , $I_a$ is the photosynthetically active radiation and $R_d$ is the dark respiration ($R_d = \frac{A_m}{9}$). $\epsilon$ is estimated with the following Eq.( A9).

$$\epsilon = \epsilon_0 \cdot (\frac{C_i - \Gamma}{C_i + 2\Gamma}) \qquad (A9)$$

where $\epsilon_0$ is the maximum quantum use efficiency in $kg_{CO2}$ $J^{-1}$ PAR

**Appendix B:**

**TEB-GREENROOF input parameters :**

**Appendix C:**

**Forced A-gs model**

In order to reduce the cost of calculation, the A-gs component of the ISBA-A-gs model has been rewritten and implemented in R, allowing for the rapid execution of simulations without the necessity of computing each step individually. Indeed, the separated A-gs model is forced with computed ISBA-A-gs input from a reference TEB-GREENROOF simulation with forced LAI. This approach enabled the simulation to run every time step simultaneously however, this approch removed the retroactive effect of the CO2 fluxes. The LAI monthly evolution for the reference simulation was constructed using the monthly average of the spline interpolation of the punctual estimated values of LAI on the green roof case study site.

**Appendix D**

**Pre-calibration** Before the calibration with the full TEB configuration, a pre-calibration is made in order to reduce the plausible range of values and combination of parameters for the calibration simulations. During the pre-calibration, the identified sensitive parameters ($g^*_{mes}(25)$, $\Gamma(25)$ $\epsilon_0$, $A_{m,max}(25)$, $f^*_0$ and $F2_{max}$) are modified by conducting multiple simulations on the forced A-gs model, with the same parameter range and sampling method as that used in the sensitivity analysis but only on the sensitive parameters. A total of 50000 simulations were conducted on the specified range of parameters. The intersection of the 0.1 % best simulation for the three scores of root-mean-square error (RMSE), mean absolute error (MAE), and (Coefficient of determination ($r^2$) were retained and are displayed on Fig. D1. The results showed that for the 0.1 % intersection for the three scores, the average value of $g^*_{mes}(25)$ for the simulations was 0.0019 m $s^{-1}$. Thus, it was decided to test for the two values 0.001 and 0.002 m $s^{-1}$. For $f^*_0$, the average was 0.56 but since the lower range was at 0.45, the values selected were 0.45 and 0.55. For $\Gamma(25)$, the average was 43.1 ppmv and the standard deviation was 8.86 ppmv but like for $f^*_0$ the values

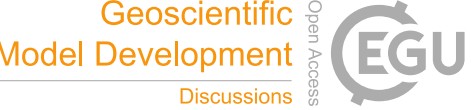

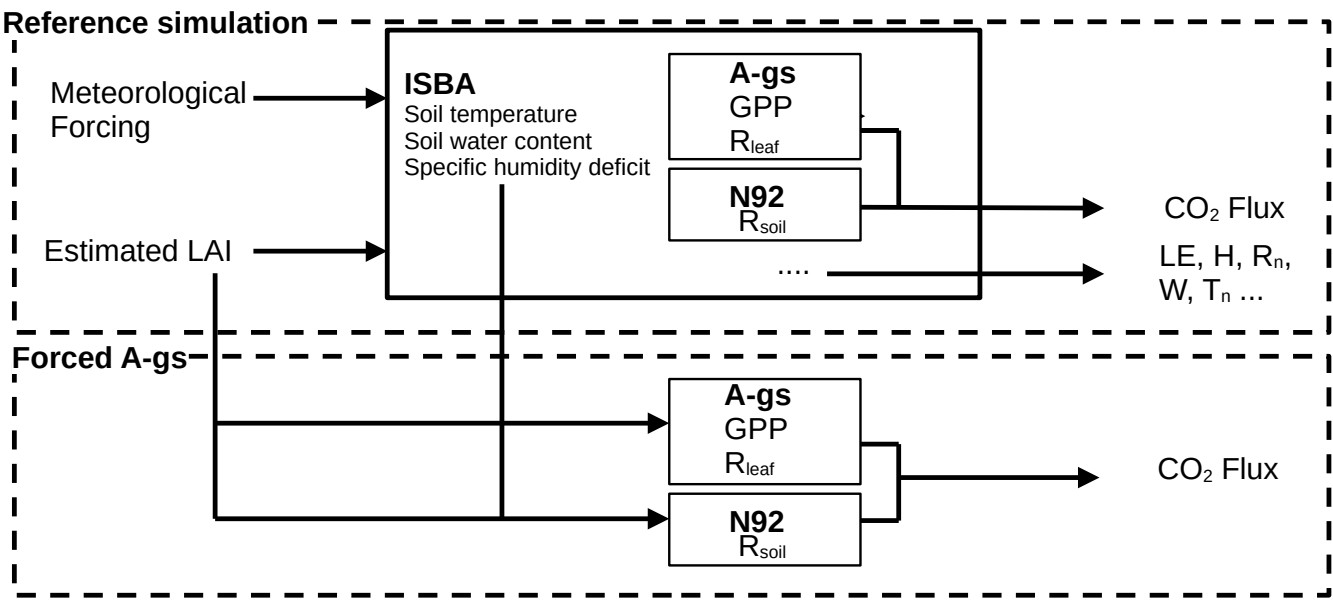

**Figure C1.** Description of the numerical setup implemented for running the A-gs photosynthesis model in a stand-alone configuration.

reached the upper range set at 49.5 ppmv so the two values selected were set to 45 ppmv and 55 ppmv. For $\epsilon_0$, $A_{m,max}(25)$ and $F2_{max}$, there was no clearly highlighted values. Consequently, the pair of values defined for $\epsilon_0$ and for $Am_{max}(25)$ were chosen based on the ISBA-a-gs C3 and C4 values. For $F2_{max}$, the values selected were 0.35 and 0.75 .

**D1**

*Author contributions.* AM did the model development, calibration and validation and wrote the paper. CM, AL, BB, VM and AL supervised
the project, gave their expertise on modelling and reviewed the paper. SW provided experimental data, gave it expertise on it and reviewed the paper.

*Competing interests.* The contact author has declared that none of the authors has any competing interests




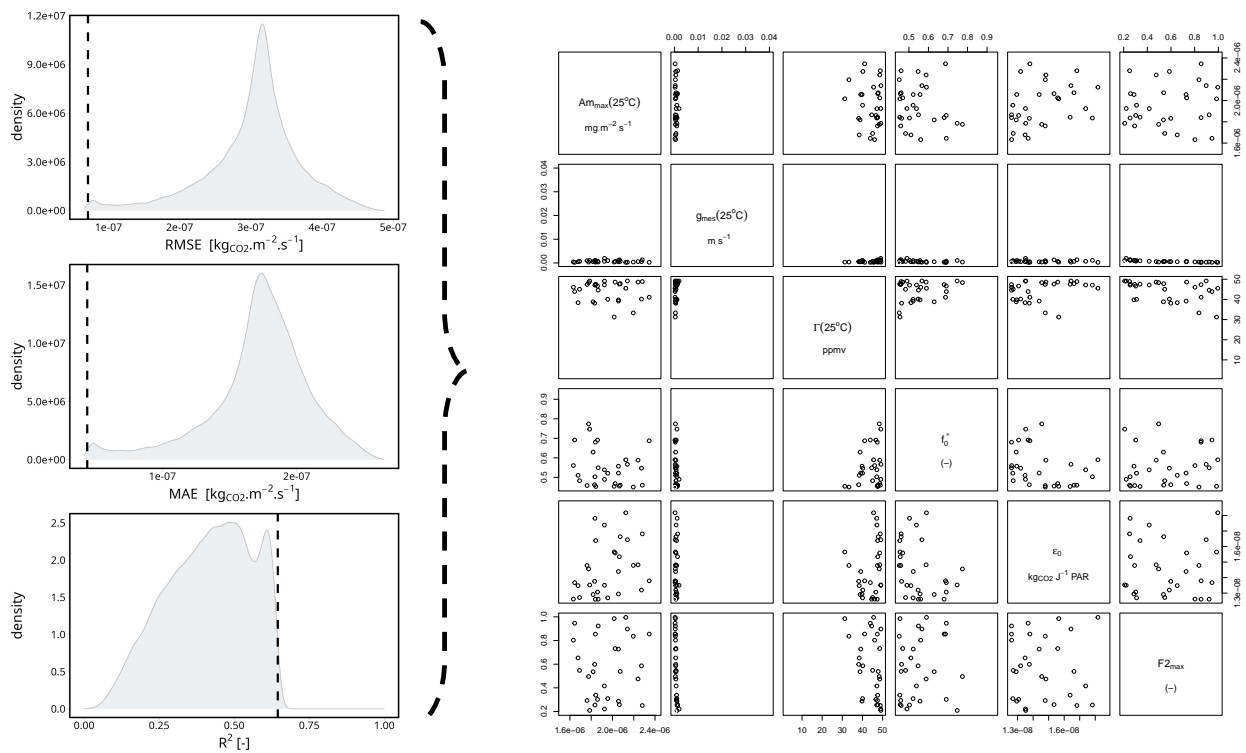

**Figure D1.** Pre-calibration results, on the left the distribution for RMSE, MAE, $r^2$

*Acknowledgements.* This research was funded by the Agence Nationale de la Recherche (ANR) under the "ANR-22-CE92-0001-01 GREEN-VELOPES" project. The thesis work of A. Mirebeau under the supervision of C. de Munck and A. Lemonsu receives a doctoral research grant from the Occitanie Region (N°00087377/21012846) and from Météo France (N°2130C0013)





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





**Table 2.** Description of the input parameters for the calculation of CO2 fluxes in TEB-GREENROOF. All the parameter values tested for the sensitivity analysis, pre-calibration and final calibration stages are listed. The data in square brackets define the ranges of values tested, and the data in brackets define the pairs of values tested.

| Parameter | Description | Unit | Sensitivity analysis / Pre-calibration | Calibration | Best config |
|---|---|---|---|---|---|
| **A-gs photosynthesis parameters** | | | | | |
| $f_0^*$ | Value of $f$ if there is no saturation deficit (with no soil water stress) | (-) | $[0.45;0.935]$ | $(0.45,0.55)$ | $0.45$ |
| $\epsilon_0$ | Maximum initial quantum use efficiency | $\mathrm{kg_{CO_2}\,J^{-1}}$ PAR | $[12.6\cdot10^{-9};18.7\cdot10^{-9}]$ | $(14\cdot10^{-9},17\cdot10^{-9})$ | $14\cdot10^{-9}$ |
| $\Gamma(25)$ | CO2 compensation concentration | ppmv | $[2.52;49.5]$ | $(2.8,55)$ | $55$ |
| $A_{m,max}(25)$ | Maximum net CO2 assimilation at 25°C | $\mathrm{mg\,m^{-2}\,s^{-1}}$ | $[1.53\cdot10^{-6};2.42\cdot10^{-6}]$ | $(1.53\cdot10^{-6},2.42\cdot10^{-6})$ | $2.2\cdot10^{-6}$ |
| $g_{mes}^*(25)$ | Mesophyll conductance at 25°C | $\mathrm{m\,s^{-1}}$ | $[10^{-3},4.0\cdot10^{-2}]$ | $(0.001,0.002)$ | $0.001$ |
| $g_c$ | Cuticular conductance | $\mathrm{m\,s^{-1}}$ | $[0;0.0001]/0.0001$ | $0.0001$ | $0.0001$ |
| $Dmax$ | Maximum saturation deficit of atmosphere tolerated by vegetation | $\mathrm{kg_{H_2O}\,kg_{air}^{-1}}$ | $[0.3;0.6]/0.3$ | $0.3$ | $0.3$ |
| **Response to drought parameters** | | | | | |
| $F2_{min}$ | Minimum normalized soil water stress factor | (-) | $0.15$ | $0.15$ | $0.15$ |
| $F2_{max}$ | Maximum normalized soil water stress factor | (-) | $[0.2;1]$ | $(0.35,0.75)$ | $0.75??$ |
| **Biomass parameters** | | | | | |
| $\tau_M$ | Maximal lifespan of leaves | days | / | $(75,150)$ | $75$ |
| $SLA$ | specific leaf area | $\mathrm{m^2\,kg\,DM^{-1}}$ | / | $12.9$ | $12.9$ |
| **Respiration parameters** | | | | | |
| $w_{10min}$ | The weighted soil volumetric water content between 0-10cm depth for respiration | (-) | $0.05$ | $0.05$ | $0.05$ |
| $w_{10max}$ | The weighted soil volumetric water content between 0-10cm depth for respiration | (-) | $0.15$ | $0.15$ | $0.15$ |
| $a$ | Coefficient for estimating vegetation coverage | (-) | / | $-0.35$ | $-0.35$ |





**Table 3.** Comparison of modelled and observed annual NEE in g C m$^{-2}$ y$^{-1}$ for each year from 2016 to 2020. By definition, a negative NEE corresponds to a quantity of carbon fixed by the green roof

|  | Calibration | | | | Evaluation |
|  | 2016 | 2017 | 2018 | 2019 | 2020 |
| Simulations | | | | | |
| --- | --- | --- | --- | --- | --- |
| NEE observed (g C m$^{-2}$ y$^{-1}$) | -163 | -178 | -85 | -151 | -168 |
| NEE modelling (g C m$^{-2}$ y$^{-1}$) | -142 | -163 | -51 | -152 | -166 |



**Table B1.** TEB-GREENROOF input parameters for road, wall and roof

| Type | Parameter | Unit | Values |
|---|---|---|---|
| **Street canyon geometry** | Building fraction | (-) | 0.6 |
| | Road fraction | (-) | 0.2 |
| | Low vegetation fraction | (-) | 0.2 |
| | Building height | m | 18 |
| | Wall density | $m^2$ wall $m^{-2}$ ground | 0.75 |
| | Low vegetation LAI | $m^2$ ; $m^{-2}$ | 2.0 |
| **Roof properties** | Number of layers | (-) | 2 |
| | Layer thickness | m | 0.05 (layer1: insulation) |
| | | | 0.16 (layer2: concrete) |
| | Roof albedo | (-) | 0.2 |
| | Roof emissivity | (-) | 0.8 |
| | Thermal conductivity | $W\ m^{-1}\ K^{-1}$ | 0.035 (layer1) |
| | | | 2.3 (layer2) |
| | Heat capacity | $kJ\ m^{-3}\ K^{-1}$ | 75 (layer1) |
| | | | 2300 (layer2) |
| **Wall properties** | Number of layers | (-) | 2 |
| | Layer thickness | m | 0.04 (layer1: concrete) |
| | | | 0.15 (layer2: concrete) |
| | Wall albedo | (-) | 0.4 |
| | Wall emissivity | (-) | 0.9 |
| | Thermal conductivity | $W\ m^{-1}\ K^{-1}$ | 2.3 (both layers) |
| | Heat capacity | $kJ\ m^{-3}\ K^{-1}$ | 2300 (both layers) |
| **Road properties** | Number of layers | (-) | 2 |
| | Layer thickness | m | 0.04 (layer1: artifical coating) |
| | | | 0.15 (layer2: soil) |
| | Road albedo | (-) | 0.08 |
| | Road emissivity | (-) | 0.94 |
| | Thermal conductivity | $W\ m^{-1}\ K^{-1}$ | 0.663 (layer1) |
| | | | 2.1 (layer2) |
| | Heat capacity | $kJ\ m^{-3}\ K^{-1}$ | 1825 (layer1) |
| | | | 2000 (layer2) |