# Peer review of "Modelling extensive green roof CO2 exchanges in the TEB urban canopy model"

_Geoscientific Model Development, 2024_

## Author Comment (AC1)

**Paper review**

aurelien.mirebeau

March 2025

**1 Anonymous Referee #1**

This paper presents the integration of the CO2 fluxes calculations in the TEB green roof model. This manuscript is already in very good conditions regarding its model implementation and testing and should be accepted after the subsequent revisions which are still considered substantial. In particular, more information should be provided on the implications of choices made in the parameterisation of the green roof scheme. The discussion section should be entirely rewritten and more comparison to the existing literature should be perpetrated. More detailed comments can be found below.

1. Regarding the introduction, you should provide more background on the different green roofs models that exist in the urban climate modelling field. You should also provide more background on the variety of applications that these models can have, list the ones that do have a CO2 implementation and comment on how the TEB model compares to other ones previously used for other types of studies. We need to better understand why you implement this CO2 fluxes scheme in TEB more than in another existing model.
The comparison to other existing model was added in the discussion part. Since there is no model that estimate CO2 fluxes with dynamic vegetation it was decided to make the comparison based on the vegetation representation. The use of the TEB model is descibed in line 36-41 (pre-review version)

2. Please define the two types of upwelling and downwelling radiations for non expert audiences.
Modified in the manuscript:
"The site is also equipped with radiometers on top of the green roof horizontally to measure both downwelling (receive by the green roof) and upwelling (reflected and emitted by the green roof) components of long-wave ($LW^\downarrow$, $LW^\uparrow$) and short-wave ($SW^\downarrow$, $SW^\uparrow$) radiation"

3. Could you please detail slightly more what sort of photograph analysis is performed ? Does it compare to other remote sensing techniques for flagging green roofs or quantifying LAI? Why is this useful for your moelling and what are the implications of potential uncertainties?

This work was made by the Technische Universität Braunschweig and further explained in the paper of Heusinger and Weber, 2017, Modified in the manuscript: "For details the reader is referred to (Heusinger and Weber, 2017b)".

4. Explain the choice for using TN and TX naming. Can you use more
The TN and TX namings for minimum and maximum daily temperatures, respectively, are the standard notations used in climatology and climate. The definition is given both in the text and in the caption of Figure 2.

Figure 1. The area covered by the green roof seems very large. Could you provide some more information on how that green roof surface compares to other green roofs in the city, in Germany and maybe even in Europe? Why does that make it an interesting case study? If it is just related to accessibility to data records, then say it. An aerial imagery would also be greatly welcomed.
The green roof is 8600 $m^2$ which at this size and climate conditions enable robust eddy covariance measurement (Heusinger and Weber, 2017a) with associated footprints representative of the surface properties of the green roof whatever the wind direction. Furthermore, no effect of anthropogenic activities on measured fluxes according to Heusinger and Weber (2017a). The site, however, is larger than the typical green roof in Berlin. The average size of the roughly 20,000 green roofs in Berlin is 287 m2 with a standard deviation of 47 ± 84 m2 (Senatsverwaltung, 2017) This makes the site a particularly interesting case study especially in order to quantify and model the carbon sequestration of green roof at wider scale.

Figure 2. Please prevent from using double Y axes. Separate the plot in two. Why do you use maximum and minimum over average temperatures? Where are these values recorded? Is it at the station on top of the roof or a local nearby AWS? If it is on top of the roof, what are the implications of the height at which records were made for your model parameterisation/evaluation (depending on how you use them)?
The figure was modified, the plot was seperated in two separated panels. The data come from the German weather station at Berlin Schönefeld airport (BSCH; ID: 00427).

Lines 94 to 107 are somewhat of a repetition to the introduction. Can't you provide more details on how these model are distinct from other existing models? What are their pros and cons? You can keep the subsequent model description with equations that is very useful. Maybe you would like to put this in the supplementary material and only relate certain equations to the calculation of CO2 fluxes which you describe later in section 3.3?
In the introduction, the aim here is to justify/explain the value of implementing a green roof module in a model such as TEB, as it enables urban climate simulations to be carried out according to different types of numerical configurations, in particular to evaluate greening scenarios from the city scale down to the

neighbourhood. However, in Section 3.1, the objective is different: to describe the concept of an urban canopy model such as TEB to describe the exchanges between urban covers and the atmosphere. It does not seem appropriate to us to merge the two or to delete one part.

Line 175. Models* Check for typos across the manuscript.
Corrected.

Lines 219 to 220. What are the implications of such recurrent forcing? I would like the authors to test for the influence on meteorological forcing since urban climate models are typically not forced every 30 minutes but more usually at 3-hourly or 6-hourly timesteps – sometimes hourly for very specific cases.
The aim of the study is to develop and evaluate the greenroof $CO_2$ flux modeling in TEB with the objective to applying it in fully-coupled urban climate simulations (TEB coupled to a weather or climate model). In this type of configuration, the surface model exchanges data with the atmosphere model according to the numerical calculation time step, which is less than 30 minutes. We have no real reason here in lowering the forcing time step, which could obviously degrade the simulation accuracy since we would be less able to capture the diurnal cycles of weather conditions (particularly incoming radiation) and the variability of precipitation. The 30-minute forcing applied here is fairly standard compared with previous studies, like Colas et al. (https://doi.org/10.5194/egusphere-2024-1039 and de Munck et al. (Geosci. Model Dev., 6, 1941–1960, 2013 www.geosci-model-dev.net/6/1941/2013/ doi:10.5194/gmd-6-1941-2013) both with an hourly forcing time step and Lemonsu et al. (https://doi.org/10.1175/JAMC-D-21-0067.1) with a 30-minute forcing.

Line 230. So the sensor is located at 20 meters. How is your model expected to behave in a "2-dimensional" model? Please provide information on how the temperature is interpolated to the roof level within the 6 vertical layers. If it relies on 2m air temperature, then what are the implications?
The 6-level vertical grid defined in the SBL version of TEB is used to discretise the layer of atmosphere inside the canyon (between the ground and the top of the urban canopy layer), i.e. in our case study from the ground to 19.15 m AGL. For surface/atmosphere exchange calculations at roof level, the model uses only the last level of the vertical grid, corresponding here to the forcing level at 19.15 m AGL (or 1.15 m above the roof).

Section 5.1 and 5.2. Is this sensitivity analysis really necessary for a non-expert audience? Can't you sum it up and provide the details in the appendix?
It is true that this section on sensitivity analysis is fairly technical, which is why we have tried to explain the issues and the method as clearly as possible (for a non-expert audience). Nevertheless, we believe that this sensitivity analysis, inspired by Aouade et al. (2020), is fully part of the overall strategy of the work presented here, since one hand it helped the calibration effort and on the other hand it give some insight about the more impactful parameters. The calibration methodology of using sensitivity analysis on a forced A-gs model is also a method that could inspire future studies on similar cases.

Line 340 to 341. What do you mean here? How was this information integrated in the model and how did that influence the outcome of your study? How are observations consistent with the LAI and Fveg observations? Can you discretise your observations based on these parameters (e.g., through seasonality)? We would need to better understand whether the observed LAI and Fveg served for the parameterisation of the model or simply its evaluation. In both cases we need to be provided with more information on the way in which this data was gathered. Visual inspection is not sufficient. This questions the validity of this data. Also, we need to understand in more details what are the implications of accurately modelling the LAI for the CO2 fluxes calculations. Since you have observations in Figure 7 about CO2 fluxes, why do you need to evaluate this intermediate step? If you can't maybe just say that the model is right but that a limitation is that you don't know whether it is right for the right reasons or not.

We added explanations in the manuscript (Section 3.3) to justify the advantages of running the model with an evolving LAI, in order to simulate the impact of weather conditions on the vegetation growth or decline:

"In the model, LAI is the variable that reflects the vegetation evolution (i.e. the CO2 accumulation). Making the LAI evolve dynamically during the simulation, rather than prescribing LAI values imposed on the model, enables to represent the impact of meteorological conditions on vegetation over the long term, and thus to simulate more realistic CO2 accumulation scenarios over longer periods."

We also added in the manuscript (Section 5.3):

"The final simulation (i.e. the evaluation simulation) is based on the best configuration obtained from the calibration stage with all parameters listed in Table 2. It is analysed and evaluated here in relation to observations of LAI and CO2 fluxes.

Figure 8 and its description should be moved to the results section. An explanation of the reason for this analysis should also be provided in the methods.
Modified in the manuscript

Figure 9 should also be provided in the results and not in the discussion. The discussion should discuss the results against other literature outcomes and try to explain the outcomes of this study based on current knowledge.
Modified in the manuscript

In general, the whole discussion section should be restructured and rewritten. The comparison to existing literature and models is critically lacking.
The discussion section has been modified in the manuscript to include a comparison of vegetation modelling between different green roof models.

**2   Anonymous Referee #2**

The authors calibrate and evaluate an existing greenroof model and photosynthesis module, coupled with an urban canopy model, against measurements of carbon dioxide fluxes and leaf area index at an extensive green roof. Results are generally satisfying, suggesting the model is capturing key processes adequately.

General comments

Methods are solid and results are interesting. The presentation, primarily the written text, could benefit from improvement throughout. In a number of cases it is difficult to follow or lacking specific definitions or other information.
The Discussion section appears to be another Results/Model Evaluation section, rather than a Discussion section. For example, little to no comparison with other studies, or contextualization of the current results with the literature, is performed. The paper can be strengthened by adding some discussion on the shortcomings of the current modelling/data acquisition process and the possibilities for applying the calibrated variables in city-regional scale simulations of green roofs (vs. the need to calibrate them for other locations, green roof characteristics, etc). The manuscript would be greatly strengthened by these additions.
The discussion has been rewritten in the manuscript to add more comparisons with existing models, particularly on how the green roof model represents vegetation evolution.

Specific comments

Lines 7-9: 'The parametrisation was .. to 2020' → 'Measurement data from an extensive... are used to calibrate and evaluate the parametrisation',
Modified in the manuscript:
*"Measurement data from an extensive Sedum non irrigated green roof located at the Berlin BER airport in Germany from 2016 to 2020 are used to calibrate and evaluate the parametrisation"*

Line 10: 'according to their influence': Not sure what this means,
Modified in the manuscript:
*"The five years of measurements were used to do a sensitivity analysis of the photosynthesis module parameters in order to quantify their influence on the photosynthesis"*

Line 12: 'even if' → 'although',
Modified in the manuscript

Lines 20-24: This is a very long and convoluted sentence,
The sentence has been divided into several parts for easier reading:
*"Differents types of impacts and interactions were studied: effects on building energy savings at building (Virk et al., 2015) and city scale (Wang et al., 2024) and under different climates(Ascione et al., 2013) ; effects on water runoff quantity (Zheng et al., 2021) and quality (Li and Babcock, 2014) ; interactions with plants, animals, and abiotic environment (Cook Patton and Bauerle, 2012) ; or benefits on air quality (Currie and Bass, 2008)"*

Lines 25: 'Firstly ... savings': This is partially true depending on the energy generation source,
Modified in the manuscript:
*"Firstly, an indirect reduction is due to less carbon being emitted as a result of building energy savings depending on the energy generation source"*

Lines 26-35: It seems unnecessary to list all the scientific names of the mentioned plants given later the authors were grouping them into C3 or C4 types. Is it also possible to put the amount of direct carbon sequestration into perspective, i.e., how much is considered significant or trivial?
We thought it would be interesting and relevant to give an overview of the different species that have been studied in previous projects. And to show the variability of the results.

Line 45: If the authors are using latex package, CO2 can be written as chem{CO_2},
Modified in the manuscript

Lines 44-51: Reads awkwardly
Modified in the manuscript:
*"Here we improve the existing model by adding the calculation of CO2 fluxes for green roofs"*

Figure 1: In addition to a site photo, can you also add schematics of all the instrument setups? Also: "airport" not "airprot"
We modified the "airport" spelling in the manuscript. We are not sure it is necessary to add a schematic of the complete roof instrumentation. The instruments setup is further described in Heusinger and Weber, 2017a, b; Kanopka et al., 2021. For our study, we are using CO2 measurements by EC and weather data, and for each of these we are already indicating the measurement heights in the text, which seems to us to be sufficient information.

Line 81: "contrasting" not "contrasted"
Modified in the manuscript

Line 90: '(for vegetation and natural soils, lakes, oceans etc.)' → , 'such as vegetation and natural soils (cite ISBA) and lakes (cite FLake)'
Modified in the manuscript:

*"The Town Energy Balance (TEB) model is integrated in the open-access SUR-FEX land surface modelling plateform (Masson et al., 2013) together with other dedicated surface models such as ISBA for vegetation and natural soils (Noilhan and Planton, 1989; Noilhan and Mahfouf, 1996) and FLake for inland waters (Mironov et al., 2010)."*

Line 104: In addition,,
Removed in the manuscript

Line 110: '+H' → 'H',
Modified in the manuscript

Lines 113-115: Can you also include the equations for LEG and LEV?
Modified in the manuscript (see equations (2) and (3)).

Lines 116: the following ISBA parametrisation:,
Modified in the manuscript

Line 122: 'allows for the discretisation' → 'discretizes',
Modified in the manuscript

Line 141: Explain the Kersten number or provide a reference, since it will not be common knowledge for most readers.
We added some clarification in the manuscript:
*"$K_e$ is the Kersten number (that is a dimensionless parameter representing the normalized thermal conductivity as a function of the degree of saturation only), and $\lambda_{sat\,j}$ and $\lambda_{dry\,j}$ are the saturated and dry soil thermal conductivities for layer j, respectively."*

Lines 143-145: 'Similar to the formulation of radiative, thermal and hydrological exchanges, CO2 fluxes in TEB-GREENROOF are adapted from an existing module in ISBA',
Modified in the manuscript

Lines 149: Suggest moving the formula for Reco given here to previous sentence, and including the formula for NEE here.
We modified in the manuscript: we felt appropriate to remove the formula for Reco because the principle is explained directly in the text. On the other hand, the formula for NEE was added.

Line 153: Suggest adding some description of C3 and C4 plants,
Physiological groups C3, C4 and CAM are well known to the vegetated surface community, and it was decided not to go into too much detail about the two plant types. However, a sentence has been now added to give more context. For CAM more context are given after.

Lines 155-164: This whole paragraph is rather difficult to read and a definition of CAM is also needed.

The acronym of CAM was aded in the manuscript,

Modified in the manuscript: *"Nonetheless, the flux measurements carried out on the green roof at Berlin airport did not reveal any periods with CAM behaviour, which would characterised by the absorption of $CO_2$ during the night. Consequently, the question of modelling the photosynthesis mechanisms specific to this particular species in the ISBA model (which does not include CAM behaviour like most large-scale vegetation models) was not investigated further in this study. However, it still requires adaptation of the model with a specific parametrisation to match the behaviour of Sedum on a shallow substrate."*

Section 3.3: Can you add the equation for Rleaf calculation? It may be better to add more context to the leaf respiration description by moving part of the content in Appendix A here, to provide context on why certain modelling decisions were made. Variables like $\Gamma$ and $\epsilon_0$ used in the sensitivity tests were not even mentioned in the main text.

The equation for Rleaf has been added in Appendix A. In order not to weigh down the document and to focus more on the modified equations, it was decided to keep only the modified equations in the main text, even if some calibrated parameters use unmodified equations.

Line 167: 'put the inhibition functions to 1', a little more context is needed,

Eqs. 8, 9: Define "Q10"

Added in the manuscript : "$Q_{10}$ are response function defined as proportional increase of a parameters for 10 degrees increase."

Line 175: 'model' $\rightarrow$ 'models',
Modified in the manuscript

Lines 182-183: "This empirical formulation is simpler than the one proposed by Calvet (2000) for herbaceous plants." The significance of this sentence is not clear.

Modified in the manuscript : *This empirical formulation is simpler than the one proposed by Calvet (2000) for herbaceous plants where plant are separated into two growing strategies. Defensive strategies is a parametrisation for plants that try to avoid stress by reducing evaporation through stomatal regulation and growing in well-watered conditions. Offensive strategies parametrisation is for plants with more efficient absorption of water by the roots or a faster growth cycle.*

Line 186: 'set to prevent extreme values': please include threshold values.
The values are not shown here, as they were defined during the calibration effort described in the following paragraph. The values are shown in Table 2.

Line 191: 'three different reservoirs' → 'three different reservoirs (Bi)',
modified in the manuscript

Line 195: Can you include some description of the formulation of Di and Ri?
The formula written here explains the general principle of biomass calculation,
but the actual decline and respiration depend numerically on whether the plant
is in a state of growth or in a state of senescence. To avoid going into too much
detail, we have chosen to retain the following formula.

Line 200: 'SLA' → 'SLA',
Modified in the manuscript

Line 202: You could consider removing (since it is redundant): "...which is
the case on the experimental site studied here"
Modified in the manuscript

Lines 208-210: 'For application ... (see Sect. 5).' → 'Since it is challenging
to find appropriate ecophysiological data to derive these parameter values for
Sedum, we chose to calibrate some of them in Section 5.'
Modified in the manuscript

Sections 2 and 4: I recommend adding a table with all the measured variables
and also indicate if set variables are used as model input/output to help shorten
the description in Section 4. It's probably better to combine Sections 4.1 and
4.2 and focus on adding details on model configuration that are not included in
the Tables. The coefficient $\beta_{coef}$ has not been mentioned previously and needs
a proper introduction. In general, it is not easy to track the various parameters
and variables, and how they are set or used in the various components of the
paper. Some unified way to improve the communication on this front would be
helpful.
We feel it is more appropriate to keep the two subsections as they refer to
different data and information: Section 4.1 describes the time series used as
atmospheric forcings for the TEB model and constructed from local weather
observations; Section 4.2 presents the input parameters (that remain fixed dur-
ing the simulation) of the TEB model describing the properties of the urban
canyon and the green roof. This type of description is consistent with that found
in other studies of the same type (e.g. Lemonsu et al. 2012, Redon et al. 2020).
Modified in manuscript: $\beta_{coef}$ is better introduced

Table 1: Can you add the symbols for each parameter? A roughness length of
0.03 m seems quite high (implying vegetation on the order of 30 cm in height).
Dry soil heat capacity is substantially higher than any published values I've
seen – is the soil composed more of clay or sand or...? It would help to have
references backing up some of these choices.
The vegetation and fraction coverage on the green roof are fairly heterogeneous
(see photo) with a combination of sedum but also some scattered fairly tall

herbaceous plants, which justifies the choice of a roughness of 3 cm. A previous study carried out as part of an internship (unpublished work) tested the sensitivity of the results (sensible and latent turbulent heat fluxes + soil temperature and humidity) to different prescribed roughness heights, and showed that the choice of an aerodynamic roughness of 3 cm was the best compromise.
Similarly, the heat capacity value of 2E+6 J m-3 K-1 was chosen by comparison with the value proposed in de Munck et al. (2013) of 1.342+6 J m-3 K-1 for a green roof of different composition, giving better simulated heat fluxes.
As already indicated in the text (Section 2), the substrate layer is composed of a mix of expanded shale, pumice and compost.

Line 230: 1.15 m is very close to the green roof surface for a wind speed measurement – is this for source area considerations?
The measurement height of. 1.15 m above ground level is a compromise to obtain valid data on turbulent exchange between the green roof and atmosphere and to have a turbulent flux source area that is almost entirely located within the green roof dimensions. The reader is referred to Heusinger and Weber (2017) and Konopka et al. (2021) for more details on data quality.

Lines 238-240: This is a bit vague. It doesn't seem like the BEM should have significant impacts on rooftop CO2 exchange or plant growth. However, if it does a few more details should be given related to the BEM setup.
Activating the BEM module for building energy in TEB is not crucial to the study of the green roof CO2 exhanges. To simplify the text and avoid any confusion, we propose deleting the sentence referring to BEM.

Line 254: "... on the specific characteristics of the current green roof to be simulated..."
Modified in the manuscript

Line 267: NEE has not been defined,
We changed in the manuscript NEE by "carbon uptake".

Line 274: "As for Si, ..."
Modified in the manuscript

Sections 5.1: I'm not sure if it's necessary to go into detail on the statistical operation, i.e., the Sobol index method, in the main text. It may be good enough to explain the two indices and move the operation procedure into the appendix.
The sensitivity analysis inspired by Aouade et al. (2020) is fully part of the work since one hand it helped the calibration effort and on the other hand it give some insight about the more impactful parameters so it was decided to keep it in the main text. Keeping the operation procedure helps understand why the number of simulation increase a lot when adding one more parameter to test. This justify the use of a forced A-gs model. We have tried to explain

it as clearly and concisely as possible, and we propose to keep it in the body of the text.

Line 286: 'd = 8'?
We modified the text by removing "d", we just mention: *eight parameters*

Line 293: What is the sampling matrix, and why is this the appropriate formula for the number of simulations. Please describe your methodology to be comprehensible to a general urban climate modeler.
The explanation of the formula would imply to give much more detail about the Monte Carlo method in the text. As a result, in order to remain consice, we prefer to remove this formula while keeping the total number of experiments that was run (to give a range of the number of simulations required for this method).
"The size of each sampling matrix (the number of different values for one parameter) is set to 5000 which implies a total of 50000 simulations based on the Monte Carlo method."

Line 299 'make possible' → 'make it possible',
Modified in the manuscript

Line 314: 'Tab. 2.' → '(Table 2)',
Modified in the manuscript

Line 319: Can you provide a reference to the two tested $\tau_m$ values?
Modified in the manuscript:
"The parameter $\tau_M$ is tested for the two values of 75 and 150 days, 150 being the default value in ISBA for C3 and C4 plants (Gibelin et al., 2006) and 75 the half of this value."

Line 334: "So a choice needs to be made" -> Delete, or clarify and correct grammar.
Modified in the manuscript

Section 5.3.1: Can you state again the rough temporal resolution of the estimated LAI from the photographs?
The LAI was estimated by capturing the variation of the green chromatic information (green fraction) in the RGB space of photographs taken at 10 different, randomly selected roof locations of the roof (cf. Heusinger and Weber, 2017). The photos were taken roughly at a monthly interval.

Line 352: The last sentence seems out of place.
Modified in the manuscript (erased)

Lines 359-362: Please quantify these discrepancies and report them in the text.
We complemented the section 5.3.2 by indicating some quantifications of model

biases both for winter mean values and a case of CO2 flux overestimation after rainfall:

*This is illustrated after the rain of June 13, 2020, when the daily average (between 6 and 18 UTC) of the difference between the observation and the model is 3.35, 4.17, 2.93 2.42 $\mu$ mol $m^{-2}$ $s^{-1}$ for the 14th, 15th, 16th and 17th respectively, leading to an overestimation of the flux for the 4 days after the rain event. Then this trend reverses, and the difference between observation and model is 0.89, -1.57, -2.35 $\mu$ mol $m^{-2}$ for the 18th, 19th and 20th respectively.*

Figure 8: Is it possible to condense this caption description while retaining the essential elements?
Without the full caption the Figure is hard to fully understand

Lines 375-384: The panels are discussed starting on 8c) and ending with 8a). Consider switching the order of the panels in Fig. 8 accordingly.
Modified in the manuscript

Figure 9: Is it one standard deviation? Also, can you make the zero line thinner?
The standard deviation is calculated for each mean value.

Line 387: "does not simulate the diurnal CO2 cycle" ? -¿ unclear, related to my next point?
Modified in the manuscript there was a mistake with legend, it was inverted

Lines 386-388: I think model and obs need to be switched in this sentence... e.g. see Fig. 9. The model has a diurnal amplitude during winter, not obs.
Modified in the manuscript there was a mistake with legend, it was inverted

Line 388: However
Modified in the manuscript

Lines 395-396: This is confusing to read, please rephrase.
Modified in the manuscript:
"The net ecosystem exchange (see 3.3) quantify the net sequestration of $CO_2$ if the NEE is negative, or the net $CO_2$ emissions if the NEE is positive."

Figure 10: Can you add a faded background grid to help assess the monthly time scale?
modified in the manuscript

Table 3: I'm confused by the split between calibration and evaluation period. Can you specify which set of parameters are used for the simulations in Section 6?
All parameter where calibrated on the calibration period, the parameters used for the evaluation simulation are listed in Table 2 in the column 'Best config'.

We modified the first sentence in Section 5.3. to clarify this point:

*"The final simulation (i.e. the evaluation simulation) is based on the best configuration obtained from the calibration stage with all parameters listed in Table 2. It is analysed and evaluated here in relation to observations of LAI and CO2 fluxes."*

Line 410: "this is ... NEE is rather high", not sure if I understand this statement,

We reworded the sentence to clarify:

*Inversely, the wetter year 2017 presents an observed NEE of -178 g C $m^{-2}$ $yr^{-1}$ that is substantially greater than for all the other years. This is partly seen in the modelling results, where the NEE is high and reaches -163 g C $m^{-2}$ $yr^{-1}$ in 2017, but it is in the same range than for year 2020 (-166 g C $m^{-2}$ $yr^{-1}$).*

---

## Author Response (AR2)

**Response**

**aurelien.mirebeau**

**May 2025**

**1 Anonymous Referee #1**

The authors have done a great deal of work in reframing the discussion. Figure 10 is of great added value. Nevertheless, some more information should be provided as asked before and should be added to the manuscript in a synthetic way; there is potential for development in the appendix.

My previous comment on the implication of LAI quantification for the modelling study is critical. Would there be other ways to gather the LAI if one does not have access to these photos? How uncertain is the LAI quantification? Could you run some simulations to test the implications of such uncertainty? Simply referring to the paper by Heusinger and Weber is not enough. Especially considering that the aim is to have this model fully coupled to regional climate models.

The observed LAI is only used in the pre-calibration process and to roughly evaluate the modelling. Quantifying the impact of the LAI uncertainty on modelling would imply doing sensitivity test on the calibration process itself rather than on the final calibration. However some more details can be given on the LAI measurement. Estimation using RGB photo analysis is the 'reference method' for this type of site. But there are other ways to estimate the LAI, as investigated in a masters thesis which looked at modelling the green roof LAI by a semi-empirical function (just weather data) and an optimized version of it (containing subtrate water content). The results can be seen on the Figure 1, they show that the LAI estimation for all 3 methods is quite similar.

Also, more context should be provided on how transferrable these outcomes are to other similar green roofs.

For water and energy issues, the works of de Munck et al. (2013) aimed to provide a standard configuration of extensive green roofs in order to perform city-scale green roof implementation scenarios, which was tested in the study of de Munck et al. (2018). The work presented here builds on this transferable model by adding the modelling of sedums, which are the most common type of vegetation on extensive green roofs. In addition, the experimental green roof site used here for calibration and evaluation and the one used by de Munck et al. (2013) are quite comparable and well representative of standard green roofs, making the new carbon flux modelling module relevant for the realization of

[Figure]

Figure 1: Comparison of the three LAI estimation method tested on the green roof site.

large-scale scenarios taking into account carbon fluxes from the new calibration.

de Munck, C. S., Lemonsu, A., Bouzouidja, R., Masson, V., and Claverie, R.: The GREENROOF module (v7.3) for modelling green roof hydrological and energetic performances within TEB, Geosci. Model Dev., 6, 1941–1960, https://doi.org/10.5194/gmd-6-1941-2013, 2013.

C. de Munck, A. Lemonsu, V. Masson, J. Le Bras, M. Bonhomme, Evaluating the impacts of greening scenarios on thermal comfort and energy and water consumptions for adapting Paris city to climate change, Urban Climate, Volume 23, 2018, Pages 260-286, ISSN 2212-0955, https://doi.org/10.1016/j.uclim.2017.01.003.
 Once this is addressed, the paper can be accepte

**2   Anonymous Referee #2**

General comments I thank the authors for modifying the manuscript according to my comments. However, additional clarification is needed. Specifically, the discussion section on dynamic vegetation seems a bit disorganized, can you provide some context on the implications of using other LAI as- signment approaches? Please also provide line numbers where the text is modified to facilitate a faster review process.

Specific comments 1. Lines 10-12: 'The five years of measurements were used to do a sensitivity analysis of the photosynthesis module parameters in order to quantify their influence on the photosynthesis' → 'Based on the five years of measurement data, a sensitivity analysis is conducted to quantify the significance of selected parametrisation parameters on the pho- tosynthesis process',

modified in the manuscript

2. Lines 31-36: Same as the original comment, please put the amount of direct carbon sequestration into perspective, i.e., how much is considered significant or trivial?

As explained in the paper of Heusinger et al. (2017), the direct carbon sequestration measured on the green roof is equivalent to the direct sequestration mean values of several grassland in Europe ($-150 \pm 200$ g C m$^{-1}$ year $^{-1}$ ; Gilmanov et al., 2007). In this perspective, even if the direct sequestration seems negligible in comparison to indirect sequestration (estimated at 7680, 7222, and 6393 g C m$^{-2}$ yr$^{-1}$ according to Seyedabadi et al., 2021), direct sequestration can still be investigated since it compared with the sequestration found in natural areas.

3. Figure 1: If the measurement setup information is available in other papers, then I suggest moving this figure into the appendix or supplement as it doesn't add any valuable information,

Modified in the manuscript, the Figure was moved to the appendix

4. Line 124: 'due tu' $\rightarrow$ 'due to',
Modified in the manuscript

5. Line 138: remove 'which allows for the',
modified in the manuscript

6. Line 165: add '(Rleaf from Eqs. A10-11)'
modified in the manuscript

7. Line 170: remove 'that is'
modified in the manuscript

8. Section 3.3: Please be extra clear that Equations introduced here in the main text are only the modified ones. And introduce all variables used in the sensitivity test (especially those with their equations in the appendix, $\Gamma$ and $\epsilon 0$) here or in some other places in the main text,

The modified equations were further introduce with the sentence LINE 181 'which modifies the equation for ... ' . All variables were further introduced in the sensitivity analysis section LINE 302 ' '

9. Line 189: 'see Appendix eq. A2 and A3' $\rightarrow$ 'denominators in Eqs. A2-A3',

Modified in the manuscript

10. Lines 210-213: The added sentences do not provide any constructive information, either delete 'This empirical ... faster growth cycle.' or explain the reason for such a simpler formulation after 'for herbaceous plants' on Line 210,

Modified in the manuscript, this sentence was deleted.

11. Line 218: 'set to' → 'prescribed during the calibration to'
Modified in the manuscript

12. Equation 15: Please provide a citation to the formulation of Di and Ri,
This simplified formulation can be found in the PhD thesis of Raphaël Garisoain:
"Évolution du cycle du carbone des tourbières pyrénéennes dans un contexte
dechangement climatique global : observation et modélisation". Sciences de la
Terre. Institut National Polytechnique de Toulouse - INPT, 2023. Français.
NNT : 2023INPT0124. tel-05031426. It was specified in the manuscript LINE
213 : ' *the principle can be expressed as follows (Garisoain, 2023)'*
13. Table 1: As originally suggested, please provide references to the chosen
parameter values,
The references are now added in the Table 1 description.

14. Section 5.1.1: I understand the authors' desire to keep this part in the
main text. However, perhaps not in the current form, as the text is technically
heavy and rather difficult to make sense of as a general urban modeller. Please
reorganize the information and consider adding a flowchart to facilitate the al-
gorithm description visually. A flowchart was added to the section 5.1.1.

15. Authors' response to (original) Section 5.3.1: Please add this information
to the main text,
The response was added to the manuscript LINE 362 ' *The observed LAI was
estimated by capturing the variation of the green chromatic information (green
fraction) in the RGB space from photographs taken at 10 different randomly se-
lected locations on the roof, approximately one a month (Heusinger and Weber,
2017b)'*

16. Line 321: 'eight parameters' → 'eight parameters (Table 2)'
Modified in the manuscript

17. Table 2: Please add a column to list the references for calibration ranges.
Also, what are the '??' in F 2max?
Modified in the manuscript, '??' were removed in the manuscript

18. Figure 8: 'indicate standard deviations' → 'indicate one standard devia-
tion',
Modified in the manuscript

19. Section 6.1: Consider changing the section head to 'Sensitivity in sedum
parameters'
The section head was change to 'Sedum model response to micro-meteorological
conditions' LINE 423

20. Figure 10: Consider changing the Caption to 'Evolution of the modelled

(lines) GPP with (a) volumetric water content (VWC, m3 m-3), (b) photosynthetically active radiation (PAR, W m-2), and (c) temperature (Ts, C), for the parametrisation of Sedum, ISBA C3, ISBA C4, and the comparison against selected observation (dots).' And move 'For comparison, the observations are ... values for photosynthesis' to Lines 474-475, which provides more context and details in the main text while keeping the essential information in the figure caption without repetition.
This was modified in the manuscript according to the comment

21. Continued: Not sure if I understand this part 'For each ...on the site',
The confusing sentence was removed in the manuscript

22. Line 445: 'quantify' → 'quantifies', I also suggest keeping '(NEE)' in the sentence,
modified in the manuscript

23. Line 490: 'were' → where,
modified in the manuscript

24. Line 493: 'energetic' → energy,
modified in the manuscript

25. Lines 495-496: 'But, Zhou et al. ... for long term simulation.' Not sure if I understand this.
The confusing sentence was removed in the manuscript
26. There are places with spelling and grammar errors throughout the main text and the Appendix, please check carefully.